# PRE-TRAINING MOLECULAR GRAPH REPRESENTATION WITH 3D GEOMETRY

**Shengchao Liu[1,2], Hanchen Wang[3], Weiyang Liu[3,4], Joan Lasenby[3], Hongyu Guo[5], Jian Tang[1,6,7]**
[1]Mila    [2]Université de Montréal    [3]University of Cambridge    [4]MPI for Intelligent Systems, Tübingen
[5]National Research Council Canada    [6]HEC Montréal    [7]CIFAR AI Chair

## ABSTRACT

Molecular graph representation learning is a fundamental problem in modern drug and material discovery. Molecular graphs are typically modeled by their 2D topological structures, but it has been recently discovered that 3D geometric information plays a more vital role in predicting molecular functionalities. However, the lack of 3D information in real-world scenarios has significantly impeded the learning of geometric graph representation. To cope with this challenge, we propose the Graph Multi-View Pre-training (GraphMVP) framework where self-supervised learning (SSL) is performed by leveraging the correspondence and consistency between 2D topological structures and 3D geometric views. GraphMVP effectively learns a 2D molecular graph encoder that is enhanced by richer and more discriminative 3D geometry. We further provide theoretical insights to justify the effectiveness of GraphMVP. Finally, comprehensive experiments show that GraphMVP can consistently outperform existing graph SSL methods. Code is available on GitHub.

## 1 INTRODUCTION

In recent years, drug discovery has drawn increasing interest in the machine learning community. Among many challenges therein, how to discriminatively represent a molecule with a vectorized embedding remains a fundamental yet open challenge. The underlying problem can be decomposed into two components: how to design a common latent space for molecule graphs (*i.e.*, designing a suitable encoder) and how to construct an objective function to supervise the training (*i.e.*, defining a learning target). Falling broadly into the second category, our paper studies self-supervised molecular representation learning by leveraging the consistency between 3D geometry and 2D topology.

Motivated by the prominent success of the pretraining-finetuning pipeline [17], unsupervisedly pre-trained graph neural networks for molecules yields promising performance on downstream tasks and becomes increasingly popular [42, 54, 82, 90, 103, 104]. The key to pre-training lies in finding an effective proxy task (*i.e.*, training objective) to leverage the power of large unlabeled datasets. Inspired by [54, 58, 79] that molecular properties [29, 54] can be better predicted by 3D geometry due to its encoded energy knowledge, we aim to make use of the 3D geometry of molecules in pre-training. However, the stereochemical structures are often very expensive to obtain, making such 3D geometric information scarce in downstream tasks. To address this problem, we propose the Graph Multi-View Pre-training (GraphMVP) framework, where a 2D molecule encoder is pre-trained with the knowledge of 3D geometry and then fine-tuned on downstream tasks without 3D information. Our learning paradigm, during pre-training, injects the knowledge of 3D molecular geometry to a 2D molecular graph encoder such that the downstream tasks can benefit from the implicit 3D geometric prior even if there is no 3D information available.

We attain the aforementioned goal by leveraging two pretext tasks on the 2D and 3D molecular graphs: one contrastive and one generative SSL. Contrastive SSL creates the supervised signal at an **inter-molecule** level: the 2D and 3D graph pairs are positive if they are from the same molecule, and negative otherwise; Then contrastive SSL [93] will align the positive pairs and contrast the negative pairs simultaneously. Generative SSL [38, 49, 91], on the other hand, obtains the supervised signal in an **intra-molecule** way: it learns a 2D/3D representation that can reconstruct its 3D/2D counterpart view for each molecule itself. To cope with the challenge of measuring the quality of reconstruction on molecule 2D and 3D space, we further propose a novel surrogate objective function called variation

representation reconstruction (VRR) for the generative SSL task, which can effectively measure such quality in the continuous representation space. The knowledge acquired by these two SSL tasks is complementary, so our GraphMVP framework integrates them to form a more discriminative 2D molecular graph representation. Consistent and significant performance improvements empirically validate the effectiveness of GraphMVP.

We give additional insights to justify the effectiveness of GraphMVP. First, GraphMVP is a self-supervised learning approach based on maximizing mutual information (MI) between 2D and 3D views, enabling the learnt representation to capture high-level factors [6, 7, 86] in molecule data. Second, we find that 3D molecular geometry is a form of privileged information [88, 89]. It has been proven that using privileged information in training can accelerate the speed of learning. We note that privileged information is only used in training, while it is not available in testing. This perfectly matches our intuition of pre-training molecular representation with 3D geometry.

Our contributions include (1) To our best knowledge, we are the first to incorporate the 3D geometric information into graph SSL; (2) We propose one contrastive and one generative SSL tasks for pre-training. Then we elaborate their difference and empirically validate that combining both can lead to a better representation; (3) We provide theoretical insights and case studies to justify why adding 3D geometry is beneficial; (4) We achieve the SOTA performance among all the SSL methods.

**Related work.** We briefly review the most related works here and include a more detailed summarization in Appendix A. Self-supervised learning (SSL) methods have attracted massive attention to graph applications [57, 59, 97, 99]. In general, there are roughly two categories of graph SSL: contrastive and generative, where they differ on the design of the supervised signals. Contrastive graph SSL [42, 82, 90, 103, 104] constructs the supervised signals at the **inter-graph** level and learns the representation by contrasting with other graphs, while generative graph SSL [34, 42, 43, 54] focuses on reconstructing the original graph at the **intra-graph** level. One of the most significant differences that separate our work from existing methods is that all previous methods **merely** focus on 2D molecular topology. However, for scientific tasks such as molecular property prediction, 3D geometry should be incorporated as it provides complementary and comprehensive information [58, 79]. To fill this gap, we propose GraphMVP to leverage the 3D geometry in graph self-supervised pre-training.

## 2 PRELIMINARIES

We first outline the key concepts and notations used in this work. Self-supervised learning (SSL) is based on the *view* design, where each view provides a specific aspect and modality of the data. Each molecule has two natural views: the 2D graph incorporates the topological structure defined by the adjacency, while the 3D graph can better reflect the geometry and spatial relation. From a chemical perspective, 3D geometric graphs focus on the *energy* while 2D graphs emphasize the *topological* information; thus they can be composed for learning more informative representation in GraphMVP. *Transformation* is an atomic operation in SSL that can extract specific information from each view. Next, we will briefly introduce how to represent these two views.

**2D Molecular Graph** represents molecules as 2D graphs, with atoms as nodes and bonds as edges respectively. We denote each 2D graph as $g_{2D} = (X, E)$, where $X$ is the atom attribute matrix and $E$ is the bond attribute matrix. Notice that here $E$ also includes the bond connectivity. Then we will apply a transformation function $T_{2D}$ on the topological graph. Given a 2D molecular graph $g_{2D}$, its representation $h_{2D}$ can be obtained from a *2D graph neural network (GNN)* model:

$$h_{2D} = \text{GNN-2D}(T_{2D}(g_{2D})) = \text{GNN-2D}(T_{2D}(X, E)). \tag{1}$$

**3D Molecular Graph** additionally includes spatial positions of the atoms, and they are needless to be static since atoms are in continual motion on *a potential energy surface* [4]. [1] The 3D structures at the local minima on this surface are named *conformer*. As the molecular properties are conformers ensembled [36], GraphMVP provides a novel perspective on adopting 3D conformers for learning better representation. Given a conformer $g_{3D} = (X, R)$, its representation via a *3D GNN* model is:

$$h_{3D} = \text{GNN-3D}(T_{3D}(g_{3D})) = \text{GNN-3D}(T_{3D}(X, R)), \tag{2}$$

---

[1]A more rigorous way of defining conformer is in [65]: a conformer is an isomer of a molecule that differs from another isomer by the rotation of a single bond in the molecule.

where $R$ is the 3D-coordinate matrix and $T_{3D}$ is the 3D transformation. In what follows, for notation simplicity, we use $x$ and $y$ for the 2D and 3D graphs, *i.e.*, $x \triangleq g_{2D}$ and $y \triangleq g_{3D}$. Then the latent representations are denoted as $h_x$ and $h_y$.

# 3 GRAPHMVP: GRAPH MULTI-VIEW PRE-TRAINING

Our model, termed as Graph Multi-View Pre-training (GraphMVP), conducts self-supervised learning (SSL) pre-training with 3D information. The 3D conformers encode rich information about the molecule energy and spatial structure, which are complementary to the 2D topology. Thus, applying SSL between the 2D and 3D views will provide a better 2D representation, which implicitly embeds the ensembles of energies and geometric information for molecules.

In the following, we first present an overview of GraphMVP, and then introduce two pretext tasks specialized concerning 3D conformation structures. Finally, we summarize a broader graph SSL family that prevails the 2D molecular graph representation learning with 3D geometry.

## 3.1 OVERVIEW OF GRAPHMVP

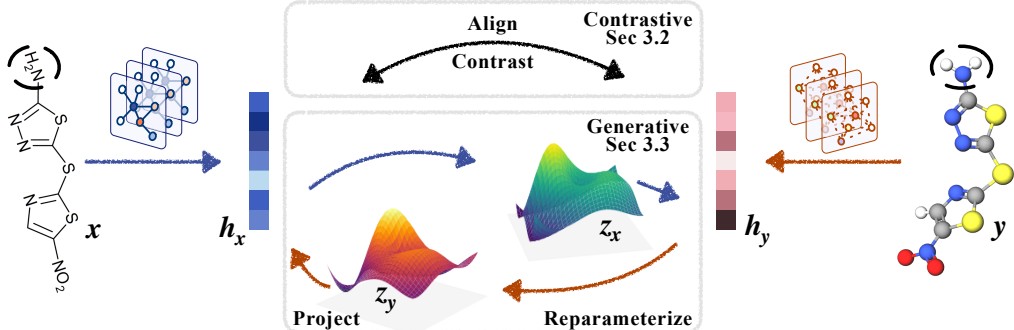

Figure 1: Overview of the pre-training stage in GraphMVP. The black dashed circles denote subgraph masking, and we mask the same region in the 2D and 3D graphs. Multiple views of the molecules (herein: Halicin) are mapped to the representation space via 2D and 3D GNN models, where we conduct GraphMVP for SSL pre-training, using both contrastive and generative pretext tasks.

In general, GraphMVP exerts 2D topology and 3D geometry as two complementary views for each molecule. By performing SSL between these views, it is expected to learn a 2D representation enhanced with 3D conformation, which can better reflect certain molecular properties.

As generic SSL pre-training pipelines, GraphMVP has two stages: pre-training then fine-tuning. In the pre-training stage, we conduct SSL via auxiliary tasks on data collections that provide both 2D and 3D molecular structures. During fine-tuning, the pre-trained 2D GNN models are subsequently fine-tuned on specific downstream tasks, where only 2D molecular graphs are available.

At the SSL pre-training stage, we design two pretext tasks: one contrastive and one generative. We conjecture and then empirically prove that these two tasks are focusing on different learning aspects, which are summarized into the following two points. (1) From the perspective of representation learning, contrastive SSL utilizes **inter-data** knowledge and generative SSL utilizes **intra-data** knowledge. For contrastive SSL, one key step is to obtain the negative view pairs for inter-data contrasting; while generative SSL focuses on each data point itself, by reconstructing the key features at an intra-data level. (2) From the perspective of distribution learning, contrastive SSL and generative SSL are learning the data distribution from a **local** and **global** manner, respectively. Contrastive SSL learns the distribution locally by contrasting the pairwise distance at an inter-data level. Thus, with sufficient number of data points, the local contrastive operation can iteratively recover the data distribution. Generative SSL, on the other hand, learns the global data density function directly.

Therefore, contrastive and generative SSL are essentially conducting representation and distribution learning with different intuitions and disciplines, and we expect that combining both can lead to a better representation. We later carry out an ablation study (Section 4.4) to verify this empirically. In

addition, to make the pretext tasks more challenging, we take views for each molecule by randomly masking $M$ nodes (and corresponding edges) as the transformation function, *i.e.*, $T_{2D} = T_{3D} = $ mask. This trick has been widely used in graph SSL [42, 103, 104] and has shown robust improvements.

## 3.2 Contrastive Self-Supervised Learning between 2D and 3D Views

The main idea of contrastive self-supervised learning (SSL) [10, 69] is first to define positive and negative pairs of views from an inter-data level, and then to align the positive pairs and contrast the negative pairs simultaneously [93]. For each molecule, we first extract representations from 2D and 3D views, *i.e.*, $h_x$ and $h_y$. Then we create positive and negative pairs for contrastive learning: the 2D-3D pairs $(x, y)$ for the same molecule are treated as positive, and negative otherwise. Finally, we align the positive pairs and contrast the negative ones. The pipeline is shown in Figure 1. In the following, we discuss two common objective functions on contrastive graph SSL.

**InfoNCE** is first proposed in [69], and its effectiveness has been validated both empirically [10, 37] and theoretically [3]. Its formulation is given as follows:

$$\mathcal{L}_{\text{InfoNCE}} = -\frac{1}{2}\mathbb{E}_{p(x,y)}\left[\log \frac{\exp(f_x(x,y))}{\exp(f_x(x,y)) + \sum_j \exp(f_x(x^j,y))}) + \log \frac{\exp(f_y(y,x))}{\exp(f_y(y,x)) + \sum_j \exp(f_y(y^j,x))}\right], \quad (3)$$

where $x^j, y^j$ are randomly sampled 2D and 3D views regarding to the anchored pair $(x, y)$. $f_x(x, y)$ and $f_y(y, x)$ are scoring functions for the two corresponding views, with flexible formulations. Here we adopt $f_x(x, y) = f_y(y, x) = \langle h_x, h_y \rangle$. More details are in Appendix D.

**Energy-Based Model with Noise Contrastive Estimation (EBM-NCE)** is an alternative that has been widely used in the line of graph contrastive SSL [42, 82, 103, 104]. Its intention is essentially the same as InfoNCE, to align positive pairs and contrast negative pairs, while the main difference is the usage of binary cross-entropy and extra noise distribution for negative sampling:

$$\mathcal{L}_{\text{EBM-NCE}} = -\frac{1}{2}\mathbb{E}_{p(y)}\left[\mathbb{E}_{p_n(x|y)}\log\left(1 - \sigma(f_x(x,y))\right) + \mathbb{E}_{p(x|y)}\log\sigma(f_x(x,y))\right]$$
$$-\frac{1}{2}\mathbb{E}_{p(x)}\left[\mathbb{E}_{p_n(y|x)}\log\left(1 - \sigma(f_y(y,x))\right) + \mathbb{E}_{p(y,x)}\log\sigma(f_y(y,x))\right], \quad (4)$$

where $p_n$ is the noise distribution and $\sigma$ is the sigmoid function. We also notice that the final formulation of EBM-NCE shares certain similarities with Jensen-Shannon estimation (JSE) [68]. However, the derivation process and underlying intuition are different: EBM-NCE models the conditional distributions in MI lower bound (Equation (9)) with EBM, while JSE is a special case of variational estimation of f-divergence. Since this is not the main focus of GraphMVP, we expand the a more comprehensive comparison in Appendix D, plus the potential benefits with EBM-NCE.

Few works [35] have witnessed the effect on the choice of objectives in graph contrastive SSL. In GraphMVP, we treat it as a hyper-parameter and further run ablation studies on them, *i.e.*, to solely use either InfoNCE ($\mathcal{L}_C = \mathcal{L}_{\text{InfoNCE}}$) or EMB-NCE ($\mathcal{L}_C = \mathcal{L}_{\text{EBM-NCE}}$).

## 3.3 Generative Self-Supervised Learning between 2D and 3D Views

Generative SSL is another classic track for unsupervised pre-training [11, 48, 49, 51]. It aims at learning an effective representation by self-reconstructing each data point. Specifically to drug discovery, we have one 2D graph and a certain number of 3D conformers for each molecule, and our goal is to learn a robust 2D/3D representation that can, to the most extent, recover its 3D/2D counterparts. By doing so, generative SSL can enforce 2D/3D GNN to encode the most crucial geometry/topology information, which can improve the downstream performance.

There are many options for generative models, including variational auto-encoder (VAE) [49], generative adversarial networks (GAN) [30], flow-based model [18], etc. In GraphMVP, we prefer VAE-like method for the following reasons: (1) The mapping between two molecular views is stochastic: multiple 3D conformers correspond to the same 2D topology; (2) An explicit 2D graph representation (*i.e.*, feature encoder) is required for downstream tasks; (3) Decoders for structured data such as graph are often highly nontrivial to design, which make them a suboptimal choice.

**Variational Molecule Reconstruction.** Therefore we propose a *light* VAE-like generative SSL, equipped with a *crafty* surrogate loss, which we describe in the following. We start with an example

for illustration. When generating 3D conformers from their corresponding 2D topology, we want to model the conditional likelihood $p(\boldsymbol{y}|\boldsymbol{x})$. By introducing a reparameterized variable $\boldsymbol{z_x} = \mu_{\boldsymbol{x}} + \sigma_{\boldsymbol{x}} \odot \epsilon$, where $\mu_{\boldsymbol{x}}$ and $\sigma_{\boldsymbol{x}}$ are two flexible functions on $h_{\boldsymbol{x}}$, $\epsilon \sim \mathcal{N}(0, I)$ and $\odot$ is the element-wise production, we have the following lower bound:

$$\log p(\boldsymbol{y}|\boldsymbol{x}) \geq \mathbb{E}_{q(\boldsymbol{z_x}|\boldsymbol{x})}\big[\log p(\boldsymbol{y}|\boldsymbol{z_x})\big] - KL(q(\boldsymbol{z_x}|\boldsymbol{x})||p(\boldsymbol{z_x})). \tag{5}$$

The expression for $\log p(\boldsymbol{x}|\boldsymbol{y})$ can be similarly derived. Equation (5) includes a conditional log-likelihood and a KL-divergence term, where the bottleneck is to calculate the first term for structured data. This term has also been recognized as the *reconstruction term*: it is essentially to reconstruct the 3D conformers ($\boldsymbol{y}$) from the sampled 2D molecular graph representation ($\boldsymbol{z_x}$). However, performing the graph reconstruction on the data space is not trivial: since molecules (*e.g.*, atoms and bonds) are discrete, modeling and measuring on the molecule space will bring extra obstacles.

**Variational Representation Reconstruction (VRR).** To cope with this challenge, we propose a novel surrogate loss by switching the reconstruction from data space to representation space. Instead of decoding the latent code $z_{\boldsymbol{x}}$ to data space, we can directly project it to the 3D representation space, denoted as $q_{\boldsymbol{x}}(z_{\boldsymbol{x}})$. Since the representation space is continuous, we may as well model the conditional log-likelihood with Gaussian distribution, resulting in L2 distance for reconstruction, *i.e.*, $\|q_{\boldsymbol{x}}(z_{\boldsymbol{x}}) - \text{SG}(h_{\boldsymbol{y}}(\boldsymbol{y}))\|^2$. Here SG is the stop-gradient operation, assuming that $h_{\boldsymbol{y}}$ is a fixed learnt representation function. SG has been widely adopted in the SSL literature to avoid model collapse [12, 31]. We call this surrogate loss as variational representation reconstruction (VRR):

$$\begin{aligned}\mathcal{L}_{\text{G}} = \mathcal{L}_{\text{VRR}} = &\frac{1}{2}\Big[\mathbb{E}_{q(\boldsymbol{z_x}|\boldsymbol{x})}\big[\|q_{\boldsymbol{x}}(\boldsymbol{z_x}) - \text{SG}(h_{\boldsymbol{y}})\|^2\big] + \mathbb{E}_{q(\boldsymbol{z_y}|\boldsymbol{y})}\big[\|q_{\boldsymbol{y}}(\boldsymbol{z_y}) - \text{SG}(h_{\boldsymbol{x}})\|_2^2\big]\Big] \\ &+ \frac{\beta}{2} \cdot \Big[KL(q(\boldsymbol{z_x}|\boldsymbol{x})||p(\boldsymbol{z_x})) + KL(q(\boldsymbol{z_y}|\boldsymbol{y})||p(\boldsymbol{z_y}))\Big].\end{aligned} \tag{6}$$

We give a simplified illustration for the generative SSL pipeline in Figure 1 and the complete derivations in Appendix E. As will be discussed in Section 5.1, VRR is actually maximizing MI, and MI is invariant to continuous bijective function [7]. Thus, this surrogate loss would be exact if the encoding function $h$ satisfies this condition. However, we find that GNN, though does not meet the condition, can provide quite robust performance, which empirically justify the effectiveness of VRR.

### 3.4 MULTI-TASK OBJECTIVE FUNCTION

As discussed before, contrastive SSL and generative SSL essentially learn the representation from distinct viewpoints. A reasonable conjecture is that combining both SSL methods can lead to overall better performance, thus we arrive at minimizing the following complete objective for GraphMVP:

$$\mathcal{L}_{\text{GraphMVP}} = \alpha_1 \cdot \mathcal{L}_{\text{C}} + \alpha_2 \cdot \mathcal{L}_{\text{G}}, \tag{7}$$

where $\alpha_1, \alpha_2$ are weighting coefficients. A later performed ablation study (Section 4.4) delivers two important messages: (1) Both individual contrastive and generative SSL on 3D conformers can consistently help improve the 2D representation learning; (2) Combining the two SSL strategies can yield further improvements. Thus, we draw the conclusion that GraphMVP (Equation (7)) is able to obtain an augmented 2D representation by fully utilizing the 3D information.

As discussed in Section 1, existing graph SSL methods only focus on the 2D topology, which is in parallel to GraphMVP: 2D graph SSL focuses on exploiting the 2D structure topology, and GraphMVP takes advantage of the 3D geometry information. Thus, we propose to merge the 2D SSL into GraphMVP. Since there are two main categories in 2D graph SSL: generative and contrastive, we propose two variants GraphMVP-G and GraphMVP-C accordingly. Their objectives are as follows:

$$\mathcal{L}_{\text{GraphMVP-G}} = \mathcal{L}_{\text{GraphMVP}} + \alpha_3 \cdot \mathcal{L}_{\text{Generative 2D-SSL}}, \quad \mathcal{L}_{\text{GraphMVP-C}} = \mathcal{L}_{\text{GraphMVP}} + \alpha_3 \cdot \mathcal{L}_{\text{Contrastive 2D-SSL}}. \tag{8}$$

Later, the empirical results also help support the effectiveness of GraphMVP-G and GraphMVP-C, and thus, we can conclude that existing 2D SSL is complementary to GraphMVP.

## 4 EXPERIMENTS AND RESULTS

### 4.1 EXPERIMENTAL SETTINGS

**Datasets.** We pre-train models on the same dataset then fine-tune on the wide range of downstream tasks. We randomly select 50k qualified molecules from GEOM [4] with both 2D and 3D structures

Table 1: Results for molecular property prediction tasks. For each downstream task, we report the mean (and standard deviation) ROC-AUC of 3 seeds with scaffold splitting. For GraphMVP, we set $M = 0.15$ and $C = 5$. The best and second best results are marked **bold** and **bold**, respectively.

| Pre-training | BBBP | Tox21 | ToxCast | Sider | ClinTox | MUV | HIV | Bace | Avg |
|---|---|---|---|---|---|---|---|---|---|
| – | 65.4(2.4) | 74.9(0.8) | 61.6(1.2) | 58.0(2.4) | 58.8(5.5) | 71.0(2.5) | 75.3(0.5) | 72.6(4.9) | 67.21 |
| EdgePred | 64.5(3.1) | 74.5(0.4) | 60.8(0.5) | 56.7(0.1) | 55.8(6.2) | 73.3(1.6) | 75.1(0.8) | 64.6(4.7) | 65.64 |
| AttrMask | 70.2(0.5) | 74.2(0.8) | 62.5(0.4) | 60.4(0.6) | 68.6(9.6) | 73.9(1.3) | 74.3(1.3) | 77.2(1.4) | 70.16 |
| GPT-GNN | 64.5(1.1) | **75.3(0.5)** | 62.2(0.1) | 57.5(4.2) | 57.8(3.1) | 76.1(2.3) | 75.1(0.2) | 77.6(0.5) | 68.27 |
| InfoGraph | 69.2(0.8) | 73.0(0.7) | 62.0(0.3) | 59.2(0.2) | 75.1(5.0) | 74.0(1.5) | 74.5(1.8) | 73.9(2.5) | 70.10 |
| ContextPred | 71.2(0.9) | 73.3(0.5) | 62.8(0.3) | 59.3(1.4) | 73.7(4.0) | 72.5(2.2) | 75.8(1.1) | 78.6(1.4) | 70.89 |
| GraphLoG | 67.8(1.7) | 73.0(0.3) | 62.2(0.4) | 57.4(2.3) | 62.0(1.8) | 73.1(1.7) | 73.4(0.6) | 78.8(0.7) | 68.47 |
| G-Contextual | 70.3(1.6) | 75.2(0.3) | 62.6(0.3) | 58.4(0.6) | 59.9(8.2) | 72.3(0.9) | 75.9(0.9) | 79.2(0.3) | 69.21 |
| G-Motif | 66.4(3.4) | 73.2(0.8) | 62.6(0.5) | 60.6(1.1) | 77.8(2.0) | 73.3(2.0) | 73.8(1.4) | 73.4(4.0) | 70.14 |
| GraphCL | 67.5(3.3) | 75.0(0.3) | 62.8(0.2) | 60.1(1.3) | 78.9(4.2) | **77.1(1.0)** | 75.0(0.4) | 68.7(7.8) | 70.64 |
| JOAO | 66.0(0.6) | 74.4(0.7) | 62.7(0.6) | 60.7(1.0) | 66.3(3.9) | 77.0(2.2) | **76.6(0.5)** | 72.9(2.0) | 69.57 |
| GraphMVP | 68.5(0.2) | 74.5(0.4) | 62.7(0.1) | **62.3(1.6)** | 79.0(2.5) | 75.0(1.4) | 74.8(1.4) | 76.8(1.1) | 71.69 |
| GraphMVP-G | 70.8(0.5) | **75.9(0.5)** | 63.1(0.2) | 60.2(1.1) | **79.1(2.8)** | **77.7(0.6)** | 76.0(0.1) | **79.3(1.5)** | **72.76** |
| GraphMVP-C | **72.4(1.6)** | 74.4(0.2) | **63.1(0.4)** | **63.9(1.2)** | 77.5(4.2) | 75.0(1.0) | **77.0(1.2)** | **81.2(0.9)** | **73.07** |

for the pre-training. As clarified in Section 3.1, conformer ensembles can better reflect the molecular property, thus we take $C$ conformers of each molecule. For downstream tasks, we first stick to the same setting of the main graph SSL work [42, 103, 104], exploring 8 binary molecular property prediction tasks, which are all in the low-data regime. Then we explore 6 regression tasks from various low-data domains to be more comprehensive. We describe all the datasets in Appendix F.

**2D GNN.** We follow the research line of SSL on molecule graph [42, 103, 104], using the same Graph Isomorphism Network (GIN) [100] as the backbone model, with the same feature sets.

**3D GNN.** We choose SchNet [79] for geometric modeling, since SchNet: (1) is found to be a strong geometric representation learning method under the fair benchmarking; (2) can be trained more efficiently, comparing to the other recent 3D models. More detailed explanations are in Appendix B.2.

## 4.2 MAIN RESULTS ON MOLECULAR PROPERTY PREDICTION.

We carry out comprehensive comparisons with 10 SSL baselines and random initialization. For pre-training, we apply all SSL methods on the same dataset based on GEOM [4]. For fine-tuning, we follow the same setting [42, 103, 104] with 8 low-data molecular property prediction tasks.

**Baselines.** Due to the rapid growth of graph SSL [59, 97, 99], we are only able to benchmark the most well-acknowledged baselines: EdgePred [34], InfoGraph [82], GPT-GNN[43], AttrMask & ContextPred[42], GraphLoG[101], G-{Contextual, Motif}[77], GraphCL[104], JOAO[103].

**Our method.** GraphMVP has two key factors: i) masking ratio ($M$) and ii) number of conformers for each molecule ($C$). We set $M = 0.15$ and $C = 5$ by default, and will explore their effects in the following ablation studies in Section 4.3. For EBM-NCE loss, we adopt the empirical distribution for noise distribution. For Equation (8), we pick the empirically optimal generative and contrastive 2D SSL method: that is AttrMask for GraphMVP-G and ContextPred for GraphMVP-C.

The main results on 8 molecular property prediction tasks are listed in Table 1. We observe that the performance of GraphMVP is significantly better than the random initialized one, and the average performance outperforms the existing SSL methods by a large margin. In addition, GraphMVP-G and GraphMVP-C consistently improve the performance, supporting the claim: **3D geometry is complementary to the 2D topology**. GraphMVP leverages the information between 3D geometry and 2D topology, and 2D SSL plays the role as regularizer to extract more 2D topological information; they are extracting different perspectives of information and are indeed complementary to each other.

## 4.3 ABLATION STUDY: THE EFFECT OF MASKING RATIO AND NUMBER OF CONFORMERS

We analyze the effects of masking ratio $M$ and the number of conformers $C$ in GraphMVP. In Table 1, we set the $M$ as 0.15 since it has been widely used in existing SSL methods [42, 103, 104], and

Table 2: Ablation of masking ratio $M$, $C \equiv 5$.

| $M$ | GraphMVP | GraphMVP-G | GraphMVP-C |
|------|----------|------------|------------|
| 0 | 71.12 | 72.15 | 72.66 |
| 0.15 | 71.60 | 72.76 | 73.08 |
| 0.30 | 71.79 | 72.91 | 73.17 |

Table 3: Ablation of # conformer $C$, $M \equiv 0.15$.

| $C$ | GraphMVP | GraphMVP-G | GraphMVP-C |
|------|----------|------------|------------|
| 1 | 71.61 | 72.80 | 72.46 |
| 5 | 71.60 | 72.76 | 73.08 |
| 10 | 72.20 | 72.59 | 73.09 |
| 20 | 72.39 | 73.00 | 73.02 |

$C$ is set to 5, which we will explain below. We explore on the range of $M \in \{0, 0.15, 0.3\}$ and $C \in \{1, 5, 10, 20\}$, and report the average performance. The complete results are in Appendix G.2.

As seen in Table 2, the improvement is more obvious from $M = 0$ (raw graph) to $M = 0.15$ than from $M = 0.15$ to $M = 0.3$. This can be explained that subgraph masking with larger ratio will make the SSL tasks more challenging, especially comparing to the raw graph ($M = 0$).

Table 3 shows the effect for $C$. We observe that the performance is generally better when adding more conformers, but will reach a plateau above certain thresholds. This observation matches with previous findings [5]: adding more conformers to augment the representation learning is not as helpful as expected; while we conclude that adding more conformers can be beneficial with little improvement. One possible reason is, when generating the dataset, we are sampling top-$C$ conformers with highest possibility and lowest energy. In other words, top-5 conformers are sufficient to cover the most conformers with equilibrium state (over 80%), and the effect of larger $C$ is thus modest.

To sum up, adding more conformers might be helpful, but the computation cost can grow linearly with the increase in dataset size. On the other hand, enlarging the masking ratio will not induce extra cost, yet the performance is slightly better. Therefore, we would encourage tuning masking ratios prior to trying a larger number of conformers from the perspective of efficiency and effectiveness.

## 4.4 ABLATION STUDY: THE EFFECT OF OBJECTIVE FUNCTION

In Section 3, we introduce a new contrastive learning objective family called EBM-NCE, and we take either InfoNCE and EBM-NCE as the contrastive SSL. For the generative SSL task, we propose a novel objective function called variational representation reconstruction (VRR) in Equation (6). As discussed in Section 3.3, stochasticity is important for GraphMVP since it can capture the conformer distribution for each 2D molecular graph. To verify this, we add an ablation study on *representation reconstruction (RR)* by removing stochasticity in VRR. Thus, here we deploy a comprehensive ablation study

Table 4: Ablation on the objective function.

| GraphMVP Loss | Contrastive | Generative | Avg |
|---------------|-------------|------------|------|
| Random | | | 67.21 |
| InfoNCE only | ✓ | | 68.85 |
| EBM-NCE only | ✓ | | 70.15 |
| VRR only | | ✓ | 69.29 |
| RR only | | ✓ | 68.89 |
| InfoNCE + VRR | ✓ | ✓ | 70.67 |
| EBM-NCE + VRR | ✓ | ✓ | 71.69 |
| InfoNCE + RR | ✓ | ✓ | 70.60 |
| EBM-NCE + RR | ✓ | ✓ | 70.94 |

to explore the effect for each individual objective function (InfoNCE, EBM-NCE, VRR and RR), followed by the pairwise combinations between them.

The results in Table 4 give certain constructive insights as follows: (1) Each individual SSL objective function (middle block) can lead to better performance. This strengthens the claim that adding 3D information is helpful for 2D representation learning. (2) According to the combination of those SSL objective functions (bottom block), adding both contrastive and generative SSL can consistently improve the performance. This verifies our claim that conducting SSL at both the inter-data and intra-data level is beneficial. (3) We can see VRR is consistently better than RR on all settings, which verifies that stochasticity is an important factor in modeling 3D conformers for molecules.

## 4.5 BROADER RANGE OF DOWNSTREAM TASKS

The 8 binary downstream tasks discussed so far have been widely applied in the graph SSL research line on molecules [42, 103, 104], but there are more tasks where the 3D conformers can be helpful. Here we test 4 extra regression property prediction tasks and 2 drug-target affinity tasks.

About the dataset statistics, more detailed information can be found in Appendix F, and we may as well briefly describe the affinity task here. Drug-target affinity (DTA) is a crucial task [70, 71, 96] in drug discovery, where it models both the molecular drugs and target proteins, with the goal to predict

Table 5: Results for four molecular property prediction tasks (regression) and two DTA tasks (regression). We report the mean RMSE of 3 seeds with scaffold splitting for molecular property downstream tasks, and mean MSE for 3 seeds with random splitting on DTA tasks. For GraphMVP, we set $M = 0.15$ and $C = 5$. The best performance for each task is marked in **bold**. We omit the std here since they are very small and indistinguishable. For complete results, please check Appendix G.4.

| Pre-training | Molecular Property Prediction | | | | | Drug-Target Affinity | | |
|---|---|---|---|---|---|---|---|---|
| | ESOL | Lipo | Malaria | CEP | Avg | Davis | KIBA | Avg |
| – | 1.178 | 0.744 | 1.127 | 1.254 | 1.0756 | 0.286 | 0.206 | 0.2459 |
| AM | 1.112 | 0.730 | 1.119 | 1.256 | 1.0542 | 0.291 | 0.203 | 0.2476 |
| CP | 1.196 | 0.702 | 1.101 | 1.243 | 1.0606 | 0.279 | 0.198 | 0.2382 |
| JOAO | 1.120 | 0.708 | 1.145 | 1.293 | 1.0663 | 0.281 | 0.196 | 0.2387 |
| GraphMVP | 1.091 | 0.718 | 1.114 | 1.236 | 1.0397 | 0.280 | 0.178 | 0.2286 |
| GraphMVP-G | 1.064 | 0.691 | 1.106 | **1.228** | 1.0221 | **0.274** | 0.175 | 0.2248 |
| GraphMVP-C | **1.029** | **0.681** | **1.097** | 1.244 | **1.0128** | 0.276 | **0.168** | **0.2223** |

their affinity scores. One recent work [66] is modeling the molecular drugs with 2D GNN and target protein (as an amino-acid sequence) with convolution neural network (CNN). We adopt this setting by pre-training the 2D GNN using GraphMVP. As illustrated in Table 5, the consistent performance gain verifies the effectiveness of our proposed GraphMVP.

## 4.6 CASE STUDY

We investigate how GraphMVP helps when the task objectives are challenging with respect to the 2D topology but straightforward using 3D geometry (as shown in Figure 2). We therefore design two case studies to testify how GraphMVP transfers knowledge from 3D geometry into the 2D representation.

The first case study is *3D Diameter Prediction*. For molecules, usually, the longer the 2D diameter is, the larger the 3D diameter (largest atomic pairwise l2 distance). However, this does not always hold, and we are interested in using the 2D graph to predict the 3D diameter. The second case study is *Long-Range Donor-Acceptor Detection*. Molecules possess a special geometric structure called donor-acceptor bond, and we want to use 2D molecular graph to detect this special structure. We validate that GraphMVP consistently brings improvements on these 2 case studies, and provide more detailed discussions and interpretations in Appendix G.6.

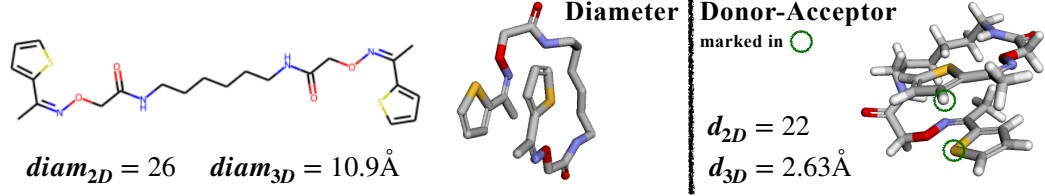

Figure 2: We select the molecules whose properties can be easily resolved via 3D but not 2D. The randomly initialised 2D GNN achieves accuracy of $38.9 \pm 0.8$ and $77.9 \pm 1.1$, respectively. The GraphMVP pre-trained ones obtain scores of $42.3 \pm 1.3$ and $81.5 \pm 0.4$, outperforming all the precedents in Section 4.2. We plot cases where random initialization fails but GraphMVP is correct.

## 5 THEORETICAL INSIGHTS

In this section, we provide the mathematical insights behind GraphMVP. We will first discuss both contrastive and generative SSL methods (Sections 3.2 and 3.3) are maximizing the mutual information (MI) and then how the 3D geometry, as privileged information, can help 2D representation learning.

### 5.1 MAXIMIZING MUTUAL INFORMATION

Mutual information (MI) measures the non-linear dependence [7] between two random variables: the larger MI, the stronger dependence between the variables. Therefore for GraphMVP, we can interpret it as maximizing MI between 2D and 3D views: to obtain a more robust 2D/3D representation

by sharing more information with its 3D/2D counterparts. This is also consistent with the sample complexity theory [3, 20, 26] where SSL as functional regularizer can reduce the uncertainty in representation learning. We first derive a lower bound for MI (see derivations in Appendix C), and the corresponding objective function $\mathcal{L}_{\text{MI}}$ is

$$I(X;Y) \geq \mathcal{L}_{\text{MI}} = \frac{1}{2}\mathbb{E}_{p(\boldsymbol{x},\boldsymbol{y})}\big[\log p(\boldsymbol{y}|\boldsymbol{x}) + \log p(\boldsymbol{x}|\boldsymbol{y})\big]. \tag{9}$$

**Contrastive Self-Supervised Learning.** InfoNCE was initialized proposed to maximize the MI directly [69]. Here in GraphMVP, EBM-NCE estimates the conditional likelihood in Equation (9) using EBM, and solves it with NCE [32]. As a result, EBM-NCE can also be seen as maximizing MI between 2D and 3D views. The detailed derivations can be found in Appendix D.2.

**Generative Self-Supervised Learning.** One alternative solution is to use a variational lower bound to approximate the conditional log-likelihood terms in Equation (9). Then we can follow the same pipeline in Section 3.3, ending up with the surrogate objective, *i.e.*, VRR in Equation (6).

## 5.2 3D GEOMETRY AS PRIVILEGED INFORMATION

We show the theoretical insights from privileged information that motivate GraphMVP. We start by considering a supervised learning setting where $(\boldsymbol{u}_i, \boldsymbol{l}_i)$ is a feature-label pair and $\boldsymbol{u}_i^*$ is the privileged information [88, 89]. The privileged information is defined to be additional information about the input $(\boldsymbol{u}_i, \boldsymbol{l}_i)$ in order to support the prediction. For example, $\boldsymbol{u}_i$ could be some CT images of a particular disease, $\boldsymbol{l}_i$ could be the label of the disease and $\boldsymbol{u}_i^*$ is the medical report from a doctor. VC theory [87, 88] characterizes the learning speed of an algorithm from the capacity of the algorithm and the amount of training data. Considering a binary classifier $f$ from a function class $\mathcal{F}$ with finite VC-dimension $\text{VCD}(\mathcal{F})$. With probability $1 - \delta$, the expected error is upper bounded by

$$R(f) \leq R_n(f) + \mathcal{O}\left(\Big(\frac{\text{VCD}(\mathcal{F}) - \log \delta}{n}\Big)^{\beta}\right) \tag{10}$$

where $R_n(f)$ denotes the training error and $n$ is the number of training samples. When the training data is separable, then $R_n(f)$ will diminish to zero and $\beta$ is equal to 1. When the training data is non-separable, $\beta$ is $\frac{1}{2}$. Therefore, the rate of convergence for the separable case is of order $1/n$. In contrast, the rate for the non-separable case is of order $1/\sqrt{n}$. We note that such a difference is huge, since the same order of bounds require up to 100 training samples versus 10,000 samples. Privileged information makes the training data separable such that the learning can be more efficient. Connecting the results to GraphMVP, we notice that the 3D geometric information of molecules can be viewed as a form of privileged information, since 3D information can effectively make molecules more separable for some properties [54, 58, 79]. Besides, privileged information is only used in training, and it well matches our usage of 3D geometry for pre-training. In fact, using 3D structures as privileged information has been already shown quite useful in protein classification [89], which serves as a strong evidence to justify the effectiveness of 3D information in graph SSL pre-training.

## 6 CONCLUSION AND FUTURE WORK

In this work, we provide a very general framework, coined GraphMVP. From the domain perspective, GraphMVP (1) is the first to incorporate 3D information for augmenting 2D graph representation learning and (2) is able to take advantages of 3D conformers by considering stochasticity in modeling. From the aspect of technical novelties, GraphMVP brings following insights when introducing 2 SSL tasks: (1) Following Equation (9), GraphMVP proposes EBM-NCE and VRR, where they are modeling the conditional distributions using EBM and variational distribution respectively. (2) EBM-NCE is similar to JSE, while we start with a different direction for theoretical intuition, yet EBM opens another promising venue in this area. (3) VRR, as a generative SSL method, is able to alleviate the potential issues in molecule generation [25, 106]. (4) Ultimately, GraphMVP combines both contrastive SSL (InfoNCE or EBM-NCE) and generative SSL (VRR) for objective function. Both empirical results (solid performance improvements on 14 downstream datasets) and theoretical analysis can strongly support the above domain and technical contributions.

We want to emphasize that GraphMVP is model-agnostic and has the potential to be expanded to many other low-data applications. This motivates broad directions for future exploration, including but not limited to: (1) More powerful 2D and 3D molecule representation methods. (2) Different application domain other than small molecules, *e.g.*, large molecules like proteins.

## REPRODUCIBILITY STATEMENT

To ensure the reproducibility of the empirical results, we include our code base in the supplementary material, which contains: instructions for installing conda virtual environment, data preprocessing scripts, and training scripts. Our code is available on GitHub for reproducibility. Complete derivations of equations and clear explanations are given in Appendices C to E.

## ACKNOWLEDGEMENT

This project is supported by the Natural Sciences and Engineering Research Council (NSERC) Discovery Grant, the Canada CIFAR AI Chair Program, collaboration grants between Microsoft Research and Mila, Samsung Electronics Co., Ltd., Amazon Faculty Research Award, Tencent AI Lab Rhino-Bird Gift Fund and a NRC Collaborative R&D Project (AI4D-CORE-06). This project was also partially funded by IVADO Fundamental Research Project grant PRF-2019-3583139727.

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

# Appendix

## Table of Contents

## A    SELF-SUPERVISED LEARNING ON MOLECULAR GRAPH

Self-supervised learning (SSL) methods have attracted massive attention recently, trending from vision [9, 10, 12, 37, 92], language [8, 17, 69] to graph [42, 54, 82, 90, 103, 104]. In general, there are two categories of SSL: contrastive and generative, where they differ on the design of the supervised signals. Contrastive SSL realizes the supervised signals at the **inter-data** level, learning the representation by contrasting with other data points; while generative SSL focuses on reconstructing the original data at the **intra-data** level. Both venues have been widely explored [57, 59, 97, 99].

### A.1    CONTRASTIVE GRAPH SSL

Contrastive graph SSL first applies transformations to construct different *views* for each graph. Each view incorporates different granularities of information, like node-, subgraph-, and graph-level. It then solves two sub-tasks simultaneously: (1) aligning the representations of views from the same data; (2) contrasting the representations of views from different data, leading to a uniformly distributed latent space [93]. The key difference among existing methods is thus the design of view constructions. InfoGraph [82, 90] contrasted the node (local) and graph (global) views. ContextPred [42] and G-Contextual [77] contrasted between node and context views. GraphCL and JOAO [103, 104] made comprehensive comparisons among four graph-level transformations and further learned to select the most effective combinations.

### A.2    GENERATIVE GRAPH SSL

Generative graph SSL aims at reconstructing important structures for each graph. By so doing, it consequently learns a representation capable of encoding key ingredients of the data. EdgePred [34] and AttrMask [42] predicted the adjacency matrix and masked tokens (nodes and edges) respectively. GPT-GNN [43] reconstructed the whole graph in an auto-regressive approach.

### A.3    PREDICTIVE GRAPH SSL

There are certain SSL methods specific to the molecular graph. For example, one central task in drug discovery is to find the important substructure or motif in molecules that can activate the target interactions. G-Motif [77] adopts domain knowledge to heuristically extract motifs for each molecule, and the SSL task is to make prediction on the existence of each motif. Different from contrastive and generative SSL, recent literature [97] takes this as predictive graph SSL, where the supervised signals are self-generated labels.

**SSL for Molecular Graphs.** Recall that all previous methods in Table 6 **merely** focus on the 2D topology. However, for science-centric tasks such as molecular property prediction, 3D geometry should be incorporated as it provides complementary and comprehensive information [58, 79]. To

Table 6: Comparison between GraphMVP and existing graph SSL methods.

| SSL Pre-training | Graph View | | SSL Category | | |
|---|---|---|---|---|---|
| | 2D Topology | 3D Geometry | Generative | Contrastive | Predictive |
| EdgePred [34] | ✓ | - | ✓ | - | - |
| AttrMask [42] | ✓ | - | ✓ | - | - |
| GPT-GNN [43] | ✓ | - | ✓ | - | - |
| InfoGraph [82, 90] | ✓ | - | - | ✓ | - |
| ContexPred [42] | ✓ | - | - | ✓ | - |
| GraphLoG [101] | ✓ | - | - | ✓ | - |
| G-Contextual [77] | ✓ | - | - | ✓ | - |
| GraphCL [104] | ✓ | - | - | ✓ | - |
| JOAO [103] | ✓ | - | - | ✓ | - |
| G-Motif [77] | ✓ | - | - | - | ✓ |
| GraphMVP (Ours) | ✓ | ✓ | ✓ | ✓ | - |

mitigate this gap, we propose GraphMVP to leverage the 3D geometry with unsupervised graph pre-training.

# B MOLECULAR GRAPH REPRESENTATION

There are two main methods for molecular graph representation learning. The first one is the molecular fingerprints. It is a hashed bit vector to describe the molecular graph. There has been re-discoveries on fingerprints-based methods [1, 44, 52, 55, 63, 75], while its has one main drawback: Random forest and XGBoost are very strong learning models on fingerprints, but they fail to take benefits of the pre-training strategy.

Graph neural network (GNN) has become another mainstream modeling methods for molecular graph representation. Existing methods can be generally split into two venues: 2D GNN and 3D GNN, depending on what levels of information is considered. 2D GNN focuses on the topological structures of the graph, like the adjacency among nodes, while 3D GNN is able to model the "energy" of molecules by taking account the spatial positions of atoms.

First, we want to highlight that GraphMVP is model-agnostic, *i.e.*, it can be applied to any 2D and 3D GNN representation function, yet the specific 2D and 3D representations are not the main focus of this work. Second, we acknowledge there are a lot of advanced 3D [23, 45, 58, 78] and 2D [13, 16, 29, 54, 100, 102] representation methods. However, considering the *graph SSL literature* and *graph representation liteature* (illustrated below), we adopt GIN [100] and SchNet [79] in current GraphMVP.

## B.1 2D MOLECULAR GRAPH NEURAL NETWORK

The 2D representation is taking each molecule as a 2D graph, with atoms as nodes and bonds as edges, *i.e.*, $g_{2D} = (X, E)$. $X \in \mathbb{R}^{n \times d_n}$ is the atom attribute matrix, where $n$ is the number of atoms (nodes) and $d_n$ is the atom attribute dimension. $E \in \mathbb{R}^{m \times d_e}$ is the bond attribute matrix, where $m$ is the number of bonds (edges) and $d_m$ is the bond attribute dimension. Notice that here $E$ also includes the connectivity. Then we will apply a transformation function $T_{2D}$ on the topological graph. Given a 2D graph $g_{2D}$, its 2D molecular representation is:

$$h_{2D} = \text{GNN-2D}(T_{2D}(g_{2D})) = \text{GNN-2D}(T_{2D}(X, E)). \tag{11}$$

The core operation of 2D GNN is the message passing function [29], which updates the node representation based on adjacency information. We have variants depending on the design of message and aggregation functions, and we pick GIN [100] in this work.

**GIN** There has been a long research line on 2D graph representation learning [13, 16, 29, 54, 100, 102]. Among these, graph isomorphism network (GIN) model [100] has been widely used as the backbone model in recent graph self-supervised learning work [42, 103, 104]. Thus, we as well adopt GIN as the base model for 2D representation.

Recall each molecule is represented as a molecular graph, *i.e.*, $g_{2D} = (X, E)$, where $X$ and $E$ are feature matrices for atoms and bonds respectively. Then the message passing function is defined as:

$$z_i^{(k+1)} = \text{MLP}_{\text{atom}}^{(k+1)} \left( z_i^{(k)} + \sum_{j \in \mathcal{N}(i)} \left( z_j^{(k)} + \text{MLP}_{\text{bond}}^{(k+1)}(E_{ij}) \right) \right), \tag{12}$$

where $z_0 = X$ and $\text{MLP}_{\text{atom}}^{(k+1)}$ and $\text{MLP}_{\text{bond}}^{(k+1)}$ are the $(l+1)$-th MLP layers on the atom- and bond-level respectively. Repeating this for $K$ times, and we can encode $K$-hop neighborhood information for each center atom in the molecular data, and we take the last layer for each node/atom representation. The graph-level molecular representation is the mean of the node representation:

$$z(\boldsymbol{x}) = \frac{1}{N} \sum_i z_i^{(K)} \tag{13}$$

## B.2 3D MOLECULAR GRAPH NEURAL NETWORK

Recently, the 3D geometric representation learning has brought breakthrough progress in molecule modeling [23, 45, 58, 78, 79]. 3D molecular graph additionally includes spatial locations of the atoms,

which needless to be static since, in real scenarios, atoms are in continual motion on *a potential energy surface* [4]. The 3D structures at the local minima on this surface are named *molecular conformation* or *conformer*. As the molecular properties are a function of the conformer ensembles [36], this reveals another limitation of existing mainstream methods: to predict properties from a single 2D or 3D graph cannot account for this fact [4], while our proposed method can alleviate this issue to a certain extent.

For specific 3D molecular graph, it additionally includes spatial positions of the atoms. We represent each conformer as $g_{3D} = (X, R)$, where $R \in \mathbb{R}^{n \times 3}$ is the 3D-coordinate matrix, and the corresponding representation is:

$$h_{3D} = \text{GNN-3D}(T_{3D}(g_{3D})) = \text{GNN-3D}(T_{3D}(X, R)), \tag{14}$$

where $R$ is the 3D-coordinate matrix and $T_{3D}$ is the 3D transformation. Note that further information such as plane and torsion angles can be solved from the positions.

**SchNet**   SchNet [79] is composed of the following key steps:

$$z_i^{(0)} = \text{embedding}(x_i)$$
$$z_i^{(t+1)} = \text{MLP}\Big( \sum_{j=1}^{n} f(x_j^{(t-1)}, r_i, r_j) \Big) \tag{15}$$
$$h_i = \text{MLP}(z_i^{(K)}),$$

where $K$ is the number of hidden layers, and

$$f(x_j, r_i, r_j) = x_j \cdot e_k(r_i - r_j) = x_j \cdot \exp(-\gamma \| \| r_i - r_j \|_2 - \mu \|_2^2) \tag{16}$$

is the continuous-filter convolution layer, enabling the modeling of continuous positions of atoms.

We adopt SchNet for the following reasons. (1) SchNet is a very strong geometric representation method after *fair* benchmarking. (2) SchNet can be trained more efficiently, comparing to the other recent 3D models. To support these two points, we make a comparison among the most recent 3D geometric models [23, 58, 78] on QM9 dataset. QM9 [98] is a molecule dataset approximating 12 thermodynamic properties calculated by density functional theory (DFT) algorithm. Notice: UNiTE [74] is the state-of-the-art 3D GNN, but it requires a commercial software for feature extraction, thus we exclude it for now.

Table 7: Reproduced MAE on QM9. 100k for training, 17,748 for val, 13,083 for test. The last column is the approximated running time.

|  | alpha | gap | homo | lumo | mu | cv | g298 | h298 | r2 | u298 | u0 | zpve | time |
|---|---|---|---|---|---|---|---|---|---|---|---|---|---|
| SchNet [79] | 0.077 | 50 | 32 | 26 | 0.030 | 0.032 | 15 | 14 | 0.122 | 14 | 14 | 1.751 | 3h |
| SE(3)-Trans [23] | 0.143 | 59 | 36 | 36 | 0.052 | 0.068 | 68 | 72 | 1.969 | 68 | 74 | 5.517 | 50h |
| EGNN [78] | 0.075 | 49 | 29 | 26 | 0.030 | 0.032 | 11 | 10 | 0.076 | 10 | 10 | 1.562 | 24h |
| SphereNet [58] | 0.054 | 41 | 22 | 19 | 0.028 | 0.027 | 10 | 8 | 0.295 | 8 | 8 | 1.401 | 50h |

Table 7 shows that, under a fair comparison (w.r.t. data splitting, seed, cuda version, etc), SchNet can reach pretty comparable performance, yet the efficiency of SchNet is much better. Combining these two points, we adopt SchNet in current version of GraphMVP.

## B.3   SUMMARY

To sum up, in GraphMVP, the most important message we want to deliver is how to design a well-motivated SSL algorithm to extract useful 3D geometry information to augment the 2D representation for downstream fine-tuning. GraphMVP is model-agnostic, and we may as well leave the more advanced 3D [23, 45, 58, 78] and 2D [13, 54, 102] GNN for future exploration.

In addition, molecular property prediction tasks have rich alternative representation methods, including SMILES [39, 95], and biological knowledge graph [56, 94]. There have been another SSL research line on them [21, 53, 107], yet they are beyond the scope of discussion in this paper.

## C   MAXIMIZE MUTUAL INFORMATION

In what follows, we will use $X$ and $Y$ to denote the data space for $2D$ graph and $3D$ graph respectively. Then the latent representations are denoted as $h_{\boldsymbol{x}}$ and $h_{\boldsymbol{y}}$.

### C.1   FORMULATION

The standard formulation for mutual information (MI) is

$$I(X;Y) = \mathbb{E}_{p(\boldsymbol{x},\boldsymbol{y})}\Big[\log\frac{p(\boldsymbol{x},\boldsymbol{y})}{p(\boldsymbol{x})p(\boldsymbol{y})}\Big]. \tag{17}$$

Another well-explained MI inspired from wikipedia is given in Figure 3.

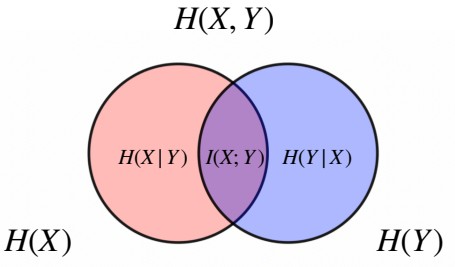

Figure 3: Venn diagram of mutual information. Inspired by wikipedia.

Mutual information (MI) between random variables measures the corresponding non-linear dependence. As can be seen in the first equation in Equation (17), the larger the divergence between the joint ($p(\boldsymbol{x},\boldsymbol{y})$ and the product of the marginals $p(\boldsymbol{x})p(\boldsymbol{y})$, the stronger the dependence between $X$ and $Y$.

Thus, following this logic, maximizing MI between 2D and 3D views can force the 3D/2D representation to capture higher-level factors, *e.g.*, the occurrence of important substructure that is semantically vital for downstream tasks. Or equivalently, maximizing MI can decrease the uncertainty in 2D representation given 3D geometric information.

### C.2   A LOWER BOUND TO MI

To solve MI, we first extract a lower bound:

$$
\begin{aligned}
I(X;Y) &= \mathbb{E}_{p(\boldsymbol{x},\boldsymbol{y})}\Big[\log\frac{p(\boldsymbol{x},\boldsymbol{y})}{p(\boldsymbol{x})p(\boldsymbol{y})}\Big] \\
&\geq \mathbb{E}_{p(\boldsymbol{x},\boldsymbol{y})}\Big[\log\frac{p(\boldsymbol{x},\boldsymbol{y})}{\sqrt{p(\boldsymbol{x})p(\boldsymbol{y})}}\Big] \\
&= \frac{1}{2}\mathbb{E}_{p(\boldsymbol{x},\boldsymbol{y})}\Big[\log\frac{(p(\boldsymbol{x},\boldsymbol{y}))^2}{p(\boldsymbol{x})p(\boldsymbol{y})}\Big] \\
&= \frac{1}{2}\mathbb{E}_{p(\boldsymbol{x},\boldsymbol{y})}\Big[\log p(\boldsymbol{x}|\boldsymbol{y})\Big] + \frac{1}{2}\mathbb{E}_{p(\boldsymbol{x},\boldsymbol{y})}\Big[\log p(\boldsymbol{y}|\boldsymbol{x})\Big] \\
&= -\frac{1}{2}[H(Y|X)+H(X|Y)].
\end{aligned} \tag{18}
$$

Thus, we transform the MI maximization problem into minimizing the following objective:

$$\mathcal{L}_{\text{MI}} = \frac{1}{2}[H(Y|X)+H(X|Y)]. \tag{19}$$

In the following sections, we will describe two self-supervised learning methods for solving MI. Notice that the methods are very general, and can be applied to various applications. Here we apply it mainly for making 3D geometry useful for 2D representation learning on molecules.

# D  CONTRASTIVE SELF-SUPERVISED LEARNING

The essence of contrastive self-supervised learning is to align positive view pairs and contrast negative view pairs, such that the obtained representation space is well distributed [93]. We display the pipeline in Figure 4. Along the research line in graph SSL [57, 59, 97, 99], InfoNCE and EBM-NCE are the two most-widely used, as discussed below.

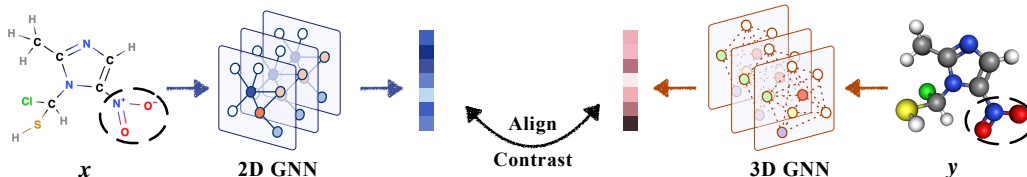

Figure 4: Contrastive SSL in GraphMVP. The black dashed circles represent subgraph masking.

## D.1  INFONCE

InfoNCE [69] is first proposed to approximate MI Equation (17):

$$\mathcal{L}_{\text{InfoNCE}} = -\frac{1}{2}\mathbb{E}\left[\log\frac{\exp(f_{\boldsymbol{x}}(\boldsymbol{x},\boldsymbol{y}))}{\exp(f_{\boldsymbol{x}}(\boldsymbol{x},\boldsymbol{y})) + \sum_j \exp(f_{\boldsymbol{x}}(\boldsymbol{x}^j,\boldsymbol{y}))}) + \log\frac{\exp(f_{\boldsymbol{y}}(\boldsymbol{y},\boldsymbol{x}))}{\exp(f_{\boldsymbol{y}}(\boldsymbol{y},\boldsymbol{x})) + \sum_j \exp f_{\boldsymbol{y}}(\boldsymbol{y}^j,\boldsymbol{x})}\right],$$
(20)

where $\boldsymbol{x}^j, \boldsymbol{y}^j$ are randomly sampled 2D and 3D views regarding to the anchored pair $(\boldsymbol{x}, \boldsymbol{y})$. $f_{\boldsymbol{x}}(\boldsymbol{x}, \boldsymbol{y}), f_{\boldsymbol{y}}(\boldsymbol{y}, \boldsymbol{x})$ are scoring functions for the two corresponding views, whose formulation can be quite flexible. Here we use $f_{\boldsymbol{x}}(\boldsymbol{x}, \boldsymbol{y}) = f_{\boldsymbol{y}}(\boldsymbol{y}, \boldsymbol{x}) = \exp(\langle h_{\boldsymbol{x}}, h_{\boldsymbol{y}}\rangle)$.

**Derivation of InfoNCE**

$$
\begin{aligned}
I(X;Y) - \log(K) &= \mathbb{E}_{p(\boldsymbol{x},\boldsymbol{y})}\Big[\log\frac{1}{K}\frac{p(\boldsymbol{x},\boldsymbol{y})}{p(\boldsymbol{x})p(\boldsymbol{y})}\Big]\\
&= \sum_{\boldsymbol{x}^i,\boldsymbol{y}^i}\Big[\log\frac{1}{K}\frac{p(\boldsymbol{x}^i,\boldsymbol{y}^i)}{p(\boldsymbol{x}^i)p(\boldsymbol{y}^i)}\Big]\\
&\geq -\sum_{\boldsymbol{x}^i,\boldsymbol{y}^i}\Big[\log\big(1 + (K-1)\frac{p(\boldsymbol{x}^i)p(\boldsymbol{y}^i)}{p(\boldsymbol{x}^i,\boldsymbol{y}^i)}\big)\Big]\\
&= -\sum_{\boldsymbol{x}^i,\boldsymbol{y}^i}\Big[\log\frac{\frac{p(\boldsymbol{x}^i,\boldsymbol{y}^i)}{p(\boldsymbol{x}^i)p(\boldsymbol{y}^i)} + (K-1)}{\frac{p(\boldsymbol{x}^i,\boldsymbol{y}^i)}{p(\boldsymbol{x}^i)p(\boldsymbol{y}^i)}}\Big]\\
&\approx -\sum_{\boldsymbol{x}^i,\boldsymbol{y}^i}\Big[\log\frac{\frac{p(\boldsymbol{x}^i,\boldsymbol{y}^i)}{p(\boldsymbol{x}^i)p(\boldsymbol{y}^i)} + (K-1)\mathbb{E}_{\boldsymbol{x}^j\neq\boldsymbol{x}^i}\frac{p(\boldsymbol{x}^j,\boldsymbol{y}^i)}{p(\boldsymbol{x}^j)p(\boldsymbol{y}^i)}}{\frac{p(\boldsymbol{x}^i,\boldsymbol{y}^i)}{p(\boldsymbol{x}^i)p(\boldsymbol{y}^i)}}\Big] \quad /\!/ \text{①}\\
&= \sum_{\boldsymbol{x}^i,\boldsymbol{y}^i}\Big[\log\frac{\exp(f_{\boldsymbol{x}}(\boldsymbol{x}^i,\boldsymbol{y}^i))}{\exp(f_{\boldsymbol{x}}(\boldsymbol{x}^i,\boldsymbol{y}^i)) + \sum_{j=1}^K f_{\boldsymbol{x}}(\boldsymbol{x}^j,\boldsymbol{y}^i)}\Big],
\end{aligned}
$$
(21)

where we set $f_{\boldsymbol{x}}(\boldsymbol{x}^i, \boldsymbol{y}^i) = \log\frac{p(\boldsymbol{x}^i,\boldsymbol{y}^i)}{p(\boldsymbol{x}^i)p(\boldsymbol{y}^i)}$.

Notice that in ①, we are using data $x \in X$ as the anchor points. If we use the $y \in Y$ as the anchor points and follow the similar steps, we can obtain

$$I(X;Y) - \log(K) \geq \sum_{\boldsymbol{y}^i,\boldsymbol{x}^i}\Big[\log\frac{\exp(f_{\boldsymbol{y}}(\boldsymbol{y}^i,\boldsymbol{x}^i))}{\exp f_{\boldsymbol{y}}(\boldsymbol{y}^i,\boldsymbol{x}^i) + \sum_{j=1}^K \exp(f_{\boldsymbol{y}}(\boldsymbol{y}^j,\boldsymbol{x}^i))}\Big].$$
(22)

Thus, by add both together, we can have the objective function as Equation (20).

## D.2 EBM-NCE

We here provide an alternative approach to maximizing MI using energy-based model (EBM). To our best knowledge, we are the **first** to give the rigorous proof of using EBM to maximize the MI.

### D.2.1 ENERGY-BASED MODEL (EBM)

Energy-based model (EBM) is a powerful tool for modeling the data distribution. The classic formulation is:

$$p(\boldsymbol{x}) = \frac{\exp(-E(\boldsymbol{x}))}{A}, \tag{23}$$

where the bottleneck is the intractable partition function $A = \int_{\boldsymbol{x}} \exp(-E(\boldsymbol{x}))d\boldsymbol{x}$. Recently, there have been quite a lot progress along this direction [19, 32, 80, 81]. Noise Contrastive Estimation (NCE) [32] is one of the powerful tools here, as we will introduce later.

### D.2.2 EBM FOR MI

Recall that our objective function is Equation (19): $\mathcal{L}_{\text{MI}} = \frac{1}{2}[H(Y|X) + H(X|Y)]$. Then we model the conditional likelihood with energy-based model (EBM). This gives us

$$\mathcal{L}_{\text{EBM}} = -\frac{1}{2}\mathbb{E}_{p(\boldsymbol{x},\boldsymbol{y})}\left[\log\frac{\exp(f_{\boldsymbol{x}}(\boldsymbol{x},\boldsymbol{y}))}{A_{\boldsymbol{x}|\boldsymbol{y}}} + \log\frac{\exp(f_{\boldsymbol{y}}(\boldsymbol{y},\boldsymbol{x}))}{A_{\boldsymbol{y}|\boldsymbol{x}}}\right], \tag{24}$$

where $f_{\boldsymbol{x}}(\boldsymbol{x},\boldsymbol{y}) = -E(\boldsymbol{x}|\boldsymbol{y})$ and $f_{\boldsymbol{y}}(\boldsymbol{y},\boldsymbol{x}) = -E(\boldsymbol{y}|\boldsymbol{x})$ are the negative energy functions, and $A_{\boldsymbol{x}|\boldsymbol{y}}$ and $A_{\boldsymbol{y}|\boldsymbol{x}}$ are the corresponding partition functions.

Under the EBM framework, if we solve Equation (24) with Noise Contrastive Estimation (NCE) [32], the final EBM-NCE objective is

$$\begin{aligned}
\mathcal{L}_{\text{EBM-NCE}} = &-\frac{1}{2}\mathbb{E}_{p_{\text{data}}(y)}\left[\mathbb{E}_{p_n(\boldsymbol{x}|\boldsymbol{y})}[\log\left(1-\sigma(f_{\boldsymbol{x}}(\boldsymbol{x},\boldsymbol{y}))\right)] + \mathbb{E}_{p_{\text{data}}(\boldsymbol{x}|\boldsymbol{y})}[\log\sigma(f_{\boldsymbol{x}}(\boldsymbol{x},\boldsymbol{y}))]\right] \\
&-\frac{1}{2}\mathbb{E}_{p_{\text{data}}(x)}\left[\mathbb{E}_{p_n(\boldsymbol{y}|\boldsymbol{x})}[\log\left(1-\sigma(f_{\boldsymbol{y}}(\boldsymbol{y},\boldsymbol{x}))\right)] + \mathbb{E}_{p_{\text{data}}(\boldsymbol{y}|\boldsymbol{x})}[\log\sigma(f_{\boldsymbol{y}}(\boldsymbol{y},\boldsymbol{x}))]\right].
\end{aligned} \tag{25}$$

Next we will give the detailed derivations.

### D.2.3 DERIVATION OF CONDITIONAL EBM WITH NCE

WLOG, let's consider the $p_\theta(\boldsymbol{x}|\boldsymbol{y})$ first, and by EBM it is as follows:

$$p_\theta(\boldsymbol{x}|\boldsymbol{y}) = \frac{\exp(-E(\boldsymbol{x}|\boldsymbol{y}))}{\int\exp(-E(\tilde{\boldsymbol{x}}|\boldsymbol{y}))d\tilde{\boldsymbol{x}}} = \frac{\exp(f_{\boldsymbol{x}}(\boldsymbol{x},\boldsymbol{y}))}{\int\exp(f_{\boldsymbol{x}}(\tilde{\boldsymbol{x}}|\boldsymbol{y}))d\tilde{\boldsymbol{x}}} = \frac{\exp(f_{\boldsymbol{x}}(\boldsymbol{x},\boldsymbol{y}))}{A_{\boldsymbol{x}|\boldsymbol{y}}}. \tag{26}$$

Then we solve this using NCE. NCE handles the intractability issue by transforming it as a binary classification task. We take the partition function $A_{\boldsymbol{x}|\boldsymbol{y}}$ as a parameter, and introduce a noise distribution $p_n$. Based on this, we introduce a mixture model: $\boldsymbol{z} = 0$ if the conditional $\boldsymbol{x}|\boldsymbol{y}$ is from $p_n(\boldsymbol{x}|\boldsymbol{y})$, and $\boldsymbol{z} = 1$ if $\boldsymbol{x}|\boldsymbol{y}$ is from $p_{\text{data}}(\boldsymbol{x}|\boldsymbol{y})$. So the joint distribution is:

$$p_{n,\text{data}}(\boldsymbol{x}|\boldsymbol{y}) = p(z=1)p_{\text{data}}(\boldsymbol{x}|\boldsymbol{y}) + p(z=0)p_n(\boldsymbol{x}|\boldsymbol{y})$$

The posterior of $p(\boldsymbol{z}=0|\boldsymbol{x},\boldsymbol{y})$ is

$$p_{n,\text{data}}(\boldsymbol{z}=0|\boldsymbol{x},\boldsymbol{y}) = \frac{p(z=0)p_n(\boldsymbol{x}|\boldsymbol{y})}{p(z=0)p_n(\boldsymbol{x}|\boldsymbol{y}) + p(z=1)p_{\text{data}}(\boldsymbol{x}|\boldsymbol{y})} = \frac{\nu \cdot p_n(\boldsymbol{x}|\boldsymbol{y})}{\nu \cdot p_n(\boldsymbol{x}|\boldsymbol{y}) + p_{\text{data}}(\boldsymbol{x}|\boldsymbol{y})},$$

where $\nu = \frac{p(\boldsymbol{z}=0)}{p(\boldsymbol{z}=1)}$.

Similarly, we can have the joint distribution under EBM framework as:

$$p_{n,\theta}(\boldsymbol{x}) = p(z=0)p_n(\boldsymbol{x}|\boldsymbol{y}) + p(z=1)p_\theta(\boldsymbol{x}|\boldsymbol{y})$$

And the corresponding posterior is:

$$p_{n,\theta}(\boldsymbol{z}=0|\boldsymbol{x},\boldsymbol{y}) = \frac{p(\boldsymbol{z}=0)p_n(\boldsymbol{x}|\boldsymbol{y})}{p(\boldsymbol{z}=0)p_n(\boldsymbol{x}|\boldsymbol{y}) + p(\boldsymbol{z}=1)p_\theta(\boldsymbol{x}|\boldsymbol{y})} = \frac{\nu \cdot p_n(\boldsymbol{x}|\boldsymbol{y})}{\nu \cdot p_n(\boldsymbol{x}|\boldsymbol{y}) + p_\theta(\boldsymbol{x}|\boldsymbol{y})}$$

We indirectly match $p_\theta(\boldsymbol{x}|\boldsymbol{y})$ to $p_{\text{data}}(\boldsymbol{x}|\boldsymbol{y})$ by fitting $p_{n,\theta}(\boldsymbol{z}|\boldsymbol{x}, \boldsymbol{y})$ to $p_{n,\text{data}}(\boldsymbol{z}|\boldsymbol{x}, \boldsymbol{y})$ by minimizing their KL-divergence:

$$
\begin{aligned}
&\min_\theta D_{\text{KL}}(p_{n,\text{data}}(\boldsymbol{z}|\boldsymbol{x}, \boldsymbol{y})||p_{n,\theta}(\boldsymbol{z}|\boldsymbol{x}, \boldsymbol{y})) \\
&= \mathbb{E}_{p_{n,\text{data}}(\boldsymbol{x},\boldsymbol{z}|\boldsymbol{y})}[\log p_{n,\theta}(\boldsymbol{z}|\boldsymbol{x}, \boldsymbol{y})] \\
&= \int \sum_{\boldsymbol{z}} p_{n,\text{data}}(\boldsymbol{x}, \boldsymbol{z}|\boldsymbol{y}) \cdot \log p_{n,\theta}(\boldsymbol{z}|\boldsymbol{x}, \boldsymbol{y})d\boldsymbol{x} \\
&= \int \Big\{ p(\boldsymbol{z}=0)p_{n,\text{data}}(\boldsymbol{x}|\boldsymbol{y}, \boldsymbol{z}=0) \log p_{n,\theta}(\boldsymbol{z}=0|\boldsymbol{x}, \boldsymbol{y}) \\
&\qquad\qquad + p(\boldsymbol{z}=1)p_{n,\text{data}}(\boldsymbol{x}|\boldsymbol{z}=1, \boldsymbol{y}) \log p_{n,\theta}(\boldsymbol{z}=1|\boldsymbol{x}, \boldsymbol{y}) \Big\}d\boldsymbol{x} \\
&= \nu \cdot \mathbb{E}_{p_n(\boldsymbol{x}|\boldsymbol{y})}\Big[ \log p_{n,\theta}(\boldsymbol{z}=0|\boldsymbol{x}, \boldsymbol{y}) \Big] + \mathbb{E}_{p_{\text{data}}(\boldsymbol{x}|\boldsymbol{y})}\Big[ \log p_{n,\theta}(\boldsymbol{z}=1|\boldsymbol{x}, \boldsymbol{y}) \Big] \\
&= \nu \cdot \mathbb{E}_{p_n(\boldsymbol{x}|\boldsymbol{y})}\Big[ \log \frac{\nu \cdot p_n(\boldsymbol{x}|\boldsymbol{y})}{\nu \cdot p_n(\boldsymbol{x}|\boldsymbol{y}) + p_\theta(\boldsymbol{x}|\boldsymbol{y})} \Big] + \mathbb{E}_{p_{\text{data}}(\boldsymbol{x}|\boldsymbol{y})}\Big[ \log \frac{p_\theta(\boldsymbol{x}|\boldsymbol{y})}{\nu \cdot p_n(\boldsymbol{x}|\boldsymbol{y}) + p_\theta(\boldsymbol{x}|\boldsymbol{y})} \Big].
\end{aligned}
\tag{27}
$$

This optimal distribution is an estimation to the actual distribution (or data distribution), *i.e.*, $p_\theta(\boldsymbol{x}|\boldsymbol{y}) \approx p_{\text{data}}(\boldsymbol{x}|\boldsymbol{y})$. We can follow the similar steps for $p_\theta(\boldsymbol{y}|\boldsymbol{x}) \approx p_{\text{data}}(\boldsymbol{y}|\boldsymbol{x})$. Thus following Equation (27), the objective function is to maximize

$$
\nu \cdot \mathbb{E}_{p_{\text{data}}(\boldsymbol{y})}\mathbb{E}_{p_n(\boldsymbol{x}|\boldsymbol{y})}\Big[ \log \frac{\nu \cdot p_n(\boldsymbol{x}|\boldsymbol{y})}{\nu \cdot p_n(\boldsymbol{x}|\boldsymbol{y}) + p_\theta(\boldsymbol{x}|\boldsymbol{y})} \Big] + \mathbb{E}_{p_{\text{data}}(\boldsymbol{y})}\mathbb{E}_{p_{\text{data}}(\boldsymbol{x}|\boldsymbol{y})}\Big[ \log \frac{p_\theta(\boldsymbol{x}|\boldsymbol{y})}{\nu \cdot p_n(\boldsymbol{x}|\boldsymbol{y}) + p_\theta(\boldsymbol{x}|\boldsymbol{y})} \Big].
\tag{28}
$$

The we will adopt three strategies to approximate Equation (28):

1. **Self-normalization.** When the EBM is very expressive, *i.e.*, using deep neural network for modeling, we can assume it is able to approximate the normalized density directly [64, 80]. In other words, we can set the partition function $A = 1$. This is a self-normalized EBM-NCE, with normalizing constant close to 1, *i.e.*, $p(\boldsymbol{x}) = \exp(-E(\boldsymbol{x})) = \exp(f(\boldsymbol{x}))$ in Equation (23).

2. **Exponential tilting term.** Exponential tilting term [2] is another useful trick. It models the distribution as $\tilde{p}_\theta(\boldsymbol{x}) = q(\boldsymbol{x})\exp(-E_\theta(\boldsymbol{x}))$, where $q(\boldsymbol{x})$ is the reference distribution. If we use the same reference distribution as the noise distribution, the tilted probability is $\tilde{p}_\theta(\boldsymbol{x}) = p_n(\boldsymbol{x})\exp(-E_\theta(\boldsymbol{x}))$ in Equation (23).

3. **Sampling.** For many cases, we only need to sample 1 negative points for each data, *i.e.*, $\nu = 1$.

Following these three disciplines, the objective function to optimize $p_\theta(\boldsymbol{x}|\boldsymbol{y})$ becomes

$$
\begin{aligned}
&\mathbb{E}_{p_n(\boldsymbol{x}|\boldsymbol{y})}\Big[ \log \frac{p_n(\boldsymbol{x}|\boldsymbol{y})}{p_n(\boldsymbol{x}|\boldsymbol{y}) + \tilde{p}_\theta(\boldsymbol{x}|\boldsymbol{y})} \Big] + \mathbb{E}_{p_{\text{data}}(\boldsymbol{x}|\boldsymbol{y})}\Big[ \log \frac{\tilde{p}_\theta(\boldsymbol{x}|\boldsymbol{y})}{p_n(\boldsymbol{x}|\boldsymbol{y}) + \tilde{p}_\theta(\boldsymbol{x}|\boldsymbol{y})} \Big] \\
&= \mathbb{E}_{p_n(\boldsymbol{x}|\boldsymbol{y})}\Big[ \log \frac{1}{1 + p_\theta(\boldsymbol{x}|\boldsymbol{y})} \Big] + \mathbb{E}_{p_{\text{data}}(\boldsymbol{x}|\boldsymbol{y})}\Big[ \log \frac{p_\theta(\boldsymbol{x}|\boldsymbol{y})}{1 + p_\theta(\boldsymbol{x}|\boldsymbol{y})} \Big] \\
&= \mathbb{E}_{p_n(\boldsymbol{x}|\boldsymbol{y})}\Big[ \log \frac{\exp(-f_{\boldsymbol{x}}(\boldsymbol{x}, \boldsymbol{y}))}{\exp(-f_{\boldsymbol{x}}(\boldsymbol{x}, \boldsymbol{y})) + 1} \Big] + \mathbb{E}_{p_{\text{data}}(\boldsymbol{x}|\boldsymbol{y})}\Big[ \log \frac{1}{\exp(-f_{\boldsymbol{x}}(\boldsymbol{x}, \boldsymbol{y})) + 1} \Big] \\
&= \mathbb{E}_{p_n(\boldsymbol{x}|\boldsymbol{y})}\Big[ \log \big(1 - \sigma(f_{\boldsymbol{x}}(\boldsymbol{x}, \boldsymbol{y}))\big) \Big] + \mathbb{E}_{p_{\text{data}}(\boldsymbol{x}|\boldsymbol{y})}\Big[ \log \sigma(f_{\boldsymbol{x}}(\boldsymbol{x}, \boldsymbol{y})) \Big].
\end{aligned}
\tag{29}
$$

Thus, the final EBM-NCE contrastive SSL objective is

$$
\begin{aligned}
\mathcal{L}_{\text{EBM-NCE}} = &-\frac{1}{2}\mathbb{E}_{p_{\text{data}}(\boldsymbol{y})}\Big[ \mathbb{E}_{p_n(\boldsymbol{x}|\boldsymbol{y})} \log \big(1 - \sigma(f_{\boldsymbol{x}}(\boldsymbol{x}, \boldsymbol{y}))\big) + \mathbb{E}_{p_{\text{data}}(\boldsymbol{x}|\boldsymbol{y})} \log \sigma(f_{\boldsymbol{x}}(\boldsymbol{x}, \boldsymbol{y})) \Big] \\
&-\frac{1}{2}\mathbb{E}_{p_{\text{data}}(\boldsymbol{x})}\Big[ \mathbb{E}_{p_n(\boldsymbol{y}|\boldsymbol{x})} \log \big(1 - \sigma(f_{\boldsymbol{y}}(\boldsymbol{y}, \boldsymbol{x}))\big) + \mathbb{E}_{p_{\text{data}}(\boldsymbol{y},\boldsymbol{x})} \log \sigma(f_{\boldsymbol{y}}(\boldsymbol{y}, \boldsymbol{x})) \Big].
\end{aligned}
\tag{30}
$$

### D.3 EBM-NCE v.s. JSE and InfoNCE

We acknowledge that there are many other contrastive objectives [73] that can be used to maximize MI. However, in the research line of graph SSL, as summarized in several recent survey papers [59, 97, 99], the two most used ones are InfoNCE and Jensen-Shannon Estimator (JSE) [40, 68].

We conclude that JSE is very similar to EBM-NCE, while the underlying perspectives are totally different, as explained below.

1. **Derivation and Intuition.** Derivation process and underlying intuition are different. JSE [68] starts from f-divergence, then with variational estimation and Fenchel duality on function $f$. Our proposed EBM-NCE is more straightforward: it models the conditional distribution in the MI lower bound Equation (19) with EBM, and solves it using NCE.

2. **Flexibility.** Modeling the conditional distribution with EBM provides a broader family of algorithms. NCE is just one solution to it, and recent progress on score matching [80, 81] and contrastive divergence [19], though no longer contrastive SSL, adds on more promising directions. Thus, EBM can provide a potential unified framework for structuring our understanding of self-supervised learning.

3. **Noise distribution.** Starting from [40], all the following works on graph SSL [59, 82, 97, 99] have been adopting the empirical distribution for noise distribution. However, this is not the case in EBM-NCE. Classic EBM-NCE uses fixed distribution, while more recent work [2] extends it with adaptively learnable noise distribution. With this discipline, more advanced sampling strategies (w.r.t. the noise distribution) can be proposed, *e.g.*, adversarial negative sampling in [41].

In the above, we conclude three key differences between EBM-NCE and JSE, plus the solid and straightforward derivations on EBM-NCE. We believe this can provide a insightful perspective of SSL to the community.

According to the empirical results Section 4.4, we observe that EBM-NCE is better than InfoNCE. This can be explained using the claim from [47], where the main technical contribution is to construct many positives and many negatives per anchor point. The binary cross-entropy in EBM-NCE is able to realize this to some extent: make all the positive pairs positive and all the negative pairs negative, where the softmax-based cross-entropy fails to capture this, as in InfoNCE.

To conclude, we are introduce using EBM in modeling MI, which opens many potential venues. As for contrastive SSL, EBM-NCE provides a better perspective than JSE, and is better than InfoNCE on graph-level self-supervised learning.

# E    GENERATIVE SELF-SUPERVISED LEARNING

Generative SSL is another classic track for unsupervised pre-training [48, 49, 51], though the main focus is on distribution learning. In GraphMVP, we start with VAE for the following reasons:

1. One of the biggest attributes of our problem is that the mapping between two views are stochastic: multiple 3D conformers can correspond to the same 2D topology. Thus, we expect a stochastic model [67] like VAE, instead of the deterministic ones.

2. For pre-training and fine-tuning, we need to learn an explicit and powerful representation function that can be used for downstream tasks.

3. The decoder for structured data like graph are often complicated, *e.g..*, the auto-regressive generation. This makes them suboptimal.

To cope with these challenges, in GraphMVP, we start with VAE-like generation model, and later propose a *light-weighted* and *smart* surrogate loss as objective function. Notice that for notation simplicity, for this section, we use $h_{\boldsymbol{y}}$ and $h_{\boldsymbol{x}}$ to delegate the 2D and 3D GNN respectively.

## E.1    VARIATIONAL MOLECULE RECONSTRUCTION

As shown in Equation (19), our main motivation is to model the conditional likelihood:

$$\mathcal{L}_{\text{MI}} = -\frac{1}{2}\mathbb{E}_{p(\boldsymbol{x},\boldsymbol{y})}[\log p(\boldsymbol{x}|\boldsymbol{y}) + \log p(\boldsymbol{y}|\boldsymbol{x})]$$

By introducing a reparameterized variable $\boldsymbol{z_x} = \mu_{\boldsymbol{x}} + \sigma_{\boldsymbol{x}} \odot \epsilon$, where $\mu_{\boldsymbol{x}}$ and $\sigma_{\boldsymbol{x}}$ are two flexible functions on $h_{\boldsymbol{x}}$, $\epsilon \sim \mathcal{N}(0, I)$ and $\odot$ is the element-wise production, we have a lower bound on the conditional likelihood:

$$\log p(\boldsymbol{y}|\boldsymbol{x}) \geq \mathbb{E}_{q(\boldsymbol{z_x}|\boldsymbol{x})}\big[\log p(\boldsymbol{y}|\boldsymbol{z_x})\big] - KL(q(\boldsymbol{z_x}|\boldsymbol{x})||p(\boldsymbol{z_x})). \tag{31}$$

Similarly, we have

$$\log p(\boldsymbol{x}|\boldsymbol{y}) \geq \mathbb{E}_{q(\boldsymbol{z_y}|\boldsymbol{y})}\big[\log p(\boldsymbol{x}|\boldsymbol{z_y})\big] - KL(q(\boldsymbol{z_y}|\boldsymbol{y})||p(\boldsymbol{z_y})), \tag{32}$$

where $\boldsymbol{z_y} = \mu_{\boldsymbol{y}} + \sigma_{\boldsymbol{y}} \odot \epsilon$. Here $\mu_{\boldsymbol{y}}$ and $\sigma_{\boldsymbol{y}}$ are flexible functions on $h_{\boldsymbol{y}}$, and $\epsilon \sim \mathcal{N}(0, I)$. For implementation, we take multi-layer perceptrons (MLPs) for $\mu_{\boldsymbol{x}}, \mu_{\boldsymbol{y}}, \sigma_{\boldsymbol{x}}, \sigma_{\boldsymbol{y}}$.

Both the above objectives are composed of a conditional log-likelihood and a KL-divergence. The conditional log-likelihood has also been recognized as the *reconstruction term*: it is essentially to reconstruct the 3D conformers ($\boldsymbol{y}$) from the sampled 2D molecular graph representation ($\boldsymbol{z_x}$). However, performing the graph reconstruction on the data space is not easy: since molecules are discrete, modeling and measuring are not trivial.

## E.2    VARIATIONAL REPRESENTATION RECONSTRUCTION

To cope with data reconstruction issue, we propose a novel generative loss termed variation representation reconstruction (VRR). The pipeline is in Figure 5.

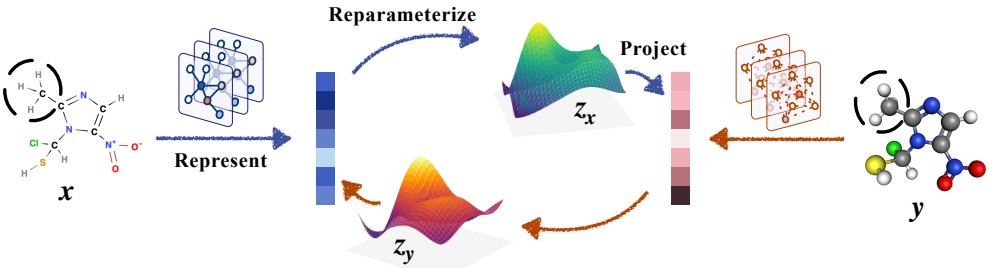

Figure 5: VRR SSL in GraphMVP. The black dashed circles represent subgraph masking.

Our proposed solution is very straightforward. Recall that MI is invariant to continuous bijective function [7]. So suppose we have a representation function $h_{\boldsymbol{y}}$ satisfying this condition, and this can guide us a surrogate loss by transferring the reconstruction from data space to the continuous representation space:

$$\mathbb{E}_{q(\boldsymbol{z_x}|\boldsymbol{x})}[\log p(\boldsymbol{y}|\boldsymbol{z_x})] = -\mathbb{E}_{q(\boldsymbol{z_x}|\boldsymbol{x})}[\|h_{\boldsymbol{y}}(g_x(\boldsymbol{z_x})) - h_{\boldsymbol{y}}(\boldsymbol{y})\|_2^2] + C,$$

where $g_x$ is the decoder and $C$ is a constant, and this introduces to using the mean-squared error (MSE) for **reconstruction on the representation space**.

Then for the reconstruction, current formula has two steps: i) the latent code $z_{\boldsymbol{x}}$ is first mapped to molecule space, and ii) it is mapped to the representation space. We can approximate these two mappings with one projection step, by directly projecting the latent code $z_{\boldsymbol{x}}$ to the 3D representation space, *i.e.*, $q_{\boldsymbol{x}}(z_{\boldsymbol{x}}) \approx h_{\boldsymbol{y}}(g_{\boldsymbol{x}}(z_{\boldsymbol{x}}))$. This gives us a variation representation reconstruction (VRR) SSL objective as below:

$$\mathbb{E}_{q(\boldsymbol{z_x}|\boldsymbol{x})}[\log p(\boldsymbol{y}|\boldsymbol{z_x})] = -\mathbb{E}_{q(\boldsymbol{z_x}|\boldsymbol{x})}[\|q_x(\boldsymbol{z_x}) - h_{\boldsymbol{y}}(\boldsymbol{y})\|_2^2] + C.$$

$\beta$-**VAE** We consider introducing a $\beta$ variable [38] to control the disentanglement of the latent representation. To be more specific, we would have

$$\log p(\boldsymbol{y}|\boldsymbol{x}) \geq \mathbb{E}_{q(\boldsymbol{z_x}|\boldsymbol{x})}\big[\log p(\boldsymbol{y}|\boldsymbol{z_x})\big] - \beta \cdot KL(q(\boldsymbol{z_x}|\boldsymbol{x})||p(\boldsymbol{z_x})). \tag{33}$$

**Stop-gradient** For the optimization on variational representation reconstruction, related work have found that adding the stop-gradient operator (SG) as a regularizer can make the training more stable without collapse both empirically [12, 31] and theoretically [84]. Here, we may as well utilize this SG operation in the objective function:

$$\mathbb{E}_{q(\boldsymbol{z_x}|\boldsymbol{x})}[\log p(\boldsymbol{y}|\boldsymbol{z_x})] = -\mathbb{E}_{q(\boldsymbol{z_x}|\boldsymbol{x})}[\|q_x(\boldsymbol{z_x}) - \text{SG}(h_{\boldsymbol{y}}(\boldsymbol{y}))\|_2^2] + C. \tag{34}$$

**Objective function for VRR** Thus, combining both two regularizers mentioned above, the final objective function for VRR is:

$$\begin{aligned}
\mathcal{L}_{\text{VRR}} = \frac{1}{2}&\Big[\mathbb{E}_{q(\boldsymbol{z_x}|\boldsymbol{x})}\big[\|q_x(\boldsymbol{z_x}) - \text{SG}(h_{\boldsymbol{y}})\|^2\big] + \mathbb{E}_{q(\boldsymbol{z_y}|\boldsymbol{y})}\big[\|q_y(\boldsymbol{z_y}) - \text{SG}(h_{\boldsymbol{x}})\|_2^2\big]\Big] \\
&+ \frac{\beta}{2} \cdot \Big[KL(q(\boldsymbol{z_x}|\boldsymbol{x})||p(\boldsymbol{z_x})) + KL(q(\boldsymbol{z_y}|\boldsymbol{y})||p(\boldsymbol{z_y}))\Big].
\end{aligned} \tag{35}$$

Note that MI is invariant to continuous bijective function [7], thus this surrogate loss would be exact if the encoding function $h_{\boldsymbol{y}}$ and $h_{\boldsymbol{x}}$ satisfy this condition. However, we find GNN (both GIN and SchNet) can, though do not meet the condition, provide quite robust performance empirically, which justify the effectiveness of VRR.

## E.3 VARIATIONAL REPRESENTATION RECONSTRUCTION AND NON-CONTRASTIVE SSL

By introducing VRR, we provide another perspective to understand the generative SSL, including the recently-proposed non-contrastive SSL [12, 31].

We provide a unified structure on the intra-data generative SSL:

- Reconstruction to the data space, like Equations (5), (31) and (32).
- Reconstruction to the representation space, *i.e.*, VRR in Equation (35).
  - If we **remove the stochasticity**, then it is simply the representation reconstruction (RR), as we tested in the ablation study Section 4.4.
  - If we **remove the stochasticity** and assume two views are **sharing the same representation function**, like CNN for multi-view learning on images, then it is reduced to the BYOL [31] and SimSiam [12]. In other words, these recently-proposed non-contrastive SSL methods are indeed special cases of VRR.

# F   DATASET OVERVIEW

## F.1   PRE-TRAINING DATASET OVERVIEW

In this section, we provide the basic statistics of the pre-training dataset (GEOM).

In Figure 6, we plot the histogram (logarithm scale on the y-axis) and cumulative distribution on the number of conformers of each molecule. As shown by the histogram and curves, there are certain number of molecules having over 1000 possible 3d conformer structures, while over 80% of molecules have less than 100 conformers.

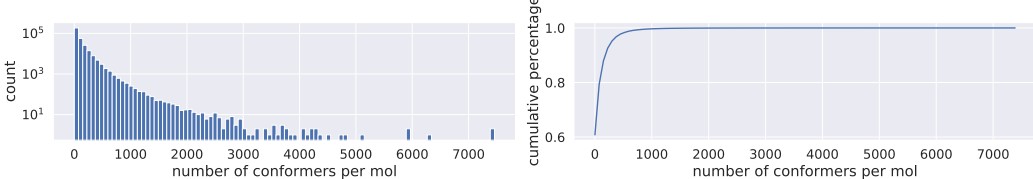

Figure 6: Statistics on the conformers of each molecule

In Figure 6, we plot the histogram of the summation of top (descending sorted by weights) {1,5,10,20} conformer weights. The physical meaning of the weight is the portion of each conformer occurred in nature. We observe that the top 5 or 10 conformers are sufficient as they have dominated nearly all the natural observations. Such long-tailed distribution is also in alignment with our findings in the ablation studies. We find that utilizing top five conformers in the GraphMVP has reached an idealised spot between effectiveness and efficiency.

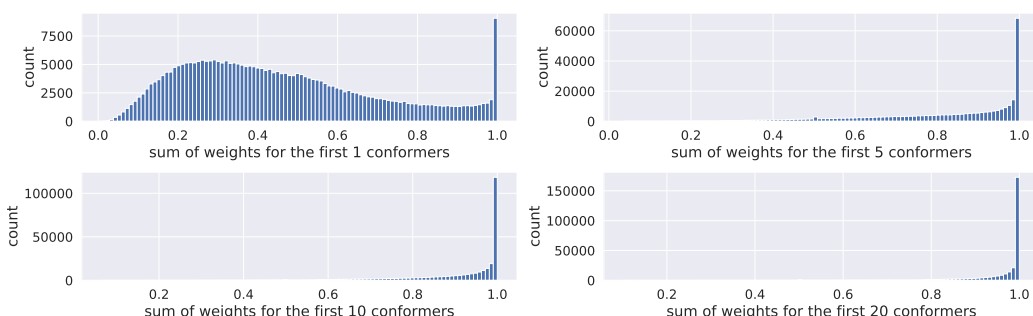

Figure 7: Sum of occurrence weights for the top major conformers

## F.2   DOWNSTREAM DATASET OVERVIEW

In this section, we review the four main categories of datasets used for downstream tasks.

**Molecular Property: Pharmacology**   The Blood-Brain Barrier Penetration (BBBP) [61] dataset measures whether a molecule will penetrate the central nervous system. All three datasets, Tox21 [85], ToxCast [98], and ClinTox [28] are related to the toxicity of molecular compounds. The Side Effect Resource (SIDER) [50] dataset stores the adverse drug reactions on a marketed drug database.

**Molecular Property: Physical Chemistry**   Dataset proposed in [15] measures aqueous solubility of the molecular compounds. Lipophilicity (Lipo) dataset is a subset of ChEMBL [27] measuring the molecule octanol/water distribution coefficient. CEP dataset is a subset of the Havard Clean Energy Project (CEP) [33], which estimates the organic photovoltaic efficiency.

**Molecular Property: Biophysics**   Maximum Unbiased Validation (MUV) [76] is another sub-database from PCBA, and is obtained by applying a refined nearest neighbor analysis. HIV is from

the Drug Therapeutics Program (DTP) AIDS Antiviral Screen [105], and it aims at predicting inhibit HIV replication. BACE measures the binding results for a set of inhibitors of $\beta$-secretase 1 (BACE-1), and is gathered in MoleculeNet [98]. Malaria [24] measures the drug efficacy against the parasite that causes malaria.

**Drug-Target Affinity**    Davis [14] measures the binding affinities between kinase inhibitors and kinases, scored by the $K_d$ value (kinase dissociation constant). KIBA [83] contains binding affinities for kinase inhibitors from different sources, including $K_i$, $K_d$ and $IC_{50}$. KIBA scores [70] are constructed to optimize the consistency among these values.

Table 8: Summary for the molecule chemical datasets.

| Dataset | Task | # Tasks | # Molecules | # Proteins | # Molecule-Protein pairs |
|---------|------|---------|-------------|------------|--------------------------|
| BBBP | Classification | 1 | 2,039 | - | - |
| Tox21 | Classification | 12 | 7,831 | - | - |
| ToxCast | Classification | 617 | 8,576 | - | - |
| Sider | Classification | 27 | 1,427 | - | - |
| ClinTox | Classification | 2 | 1,478 | - | - |
| MUV | Classification | 17 | 93,087 | - | - |
| HIV | Classification | 1 | 41,127 | - | - |
| Bace | Classification | 1 | 1,513 | - | - |
| Delaney | Regression | 1 | 1,128 | - | - |
| Lipo | Regression | 1 | 4,200 | - | - |
| Malaria | Regression | 1 | 9,999 | - | - |
| CEP | Regression | 1 | 29,978 | - | - |
| Davis | Regression | 1 | 68 | 379 | 30,056 |
| KIBA | Regression | 1 | 2,068 | 229 | 118,254 |

# G    EXPERIMENTS DETAILS

## G.1    SELF-SUPERVISED LEARNING BASELINES

For the SSL baselines in main results (Table 1), generally we can match with the original paper, even though most of them are using larger pre-training datasets, like ZINC-2m. Yet, we would like to add some specifications.

- G-{Contextual, Motif}[77] proposes a new GNN model for backbone model, and does pre-training on a larger dataset. Both settings are different from us.
- JOAO [103] has two versions in the original paper. In this paper, we run both versions and report the optimal one.
- Almost all the graph SSL baselines are reporting the test performance with optimal validation error, while GraphLoG [101] reports 73.2 in the paper with the last-epoch performance. This can be over-optimized in terms of overfitting, and here we rerun it with the same downstream evaluation strategy as a fair comparison.

## G.2 ABLATION STUDY: THE EFFECT OF MASKING RATIO AND NUMBER OF CONFORMERS

Table 9: Full results for ablation of masking ratio $M$ ($C = 0.15$), MVP is short for GraphMVP.

|  | $M$ | BBBP | Tox21 | ToxCast | Sider | ClinTox | MUV | HIV | Bace | Avg |
|---|---|---|---|---|---|---|---|---|---|---|
| – | – | 65.4(2.4) | 74.9(0.8) | 61.6(1.2) | 58.0(2.4) | 58.8(5.5) | 71.0(2.5) | 75.3(0.5) | 72.6(4.9) | 67.21 |
| MVP | 0 | 69.4 (1.0) | 75.3 (0.5) | 62.8 (0.2) | 61.9 (0.5) | 74.4 (1.3) | 74.6 (1.4) | 74.6 (1.0) | 76.0 (2.0) | 71.12 |
|  | 0.15 | 68.5 (0.2) | 74.5 (0.4) | 62.7 (0.1) | 62.3 (1.6) | 79.0 (2.5) | 75.0 (1.4) | 74.8 (1.4) | 76.8 (1.1) | 71.69 |
|  | 0.3 | 68.6 (0.3) | 74.9 (0.6) | 62.8 (0.4) | 60.0 (0.6) | 74.8 (7.8) | 74.7 (0.8) | 75.5 (1.1) | 82.9 (1.7) | 71.79 |
| MVP-G | 0 | 72.4 (1.3) | 74.7 (0.6) | 62.4 (0.2) | 60.3 (0.7) | 76.2 (5.7) | 76.6 (1.7) | 76.4 (1.7) | 78.0 (1.1) | 72.15 |
|  | 0.15 | 70.8 (0.5) | 75.9 (0.5) | 63.1 (0.2) | 60.2 (1.1) | 79.1 (2.8) | 77.7 (0.6) | 76.0 (0.1) | 79.3 (1.5) | 72.76 |
|  | 0.3 | 69.5 (0.5) | 74.6 (0.6) | 62.7 (0.3) | 60.8 (1.2) | 80.7 (2.0) | 77.8 (2.5) | 76.2 (0.5) | 81.0 (1.0) | 72.91 |
| MVP-C | 0 | 71.5 (0.9) | 75.4 (0.3) | 63.6 (0.5) | 61.8 (0.6) | 77.3 (1.2) | 75.8 (0.6) | 76.1 (0.9) | 79.8 (0.4) | 72.66 |
|  | 0.15 | 72.4 (1.6) | 74.4 (0.2) | 63.1 (0.4) | 63.9 (1.2) | 77.5 (4.2) | 75.0 (1.0) | 77.0 (1.2) | 81.2 (0.9) | 73.07 |
|  | 0.3 | 70.7 (0.8) | 74.6 (0.3) | 63.8 (0.7) | 60.4 (0.6) | 83.5 (3.2) | 74.2 (1.6) | 76.0 (1.0) | 82.2 (2.2) | 73.17 |

Table 10: Full results for ablation of # conformers $C$ ($M = 0.5$), MVP is short for GraphMVP.

|  | $C$ | BBBP | Tox21 | ToxCast | Sider | ClinTox | MUV | HIV | Bace | Avg |
|---|---|---|---|---|---|---|---|---|---|---|
| – | – | 65.4(2.4) | 74.9(0.8) | 61.6(1.2) | 58.0(2.4) | 58.8(5.5) | 71.0(2.5) | 75.3(0.5) | 72.6(4.9) | 67.21 |
| MVP | 1 | 69.2 (1.0) | 74.7 (0.4) | 62.5 (0.2) | 63.0 (0.4) | 73.9 (7.2) | 76.2 (0.4) | 75.3 (1.1) | 78.0 (0.5) | 71.61 |
|  | 5 | 68.5 (0.2) | 74.5 (0.4) | 62.7 (0.1) | 62.3 (1.6) | 79.0 (2.5) | 75.0 (1.4) | 74.8 (1.4) | 76.8 (1.1) | 71.69 |
|  | 10 | 68.3 (0.5) | 74.2 (0.6) | 63.2 (0.5) | 61.4 (1.0) | 80.6 (0.8) | 75.4 (2.4) | 75.5 (0.6) | 79.1 (2.3) | 72.20 |
|  | 20 | 68.7 (0.5) | 74.9 (0.3) | 62.7 (0.3) | 60.8 (0.7) | 75.8 (0.5) | 76.3 (1.5) | 77.4 (0.3) | 82.3 (0.8) | 72.39 |
| MVP-G | 1 | 70.9 (0.4) | 75.3 (0.7) | 62.8 (0.5) | 61.2 (0.6) | 81.4 (3.7) | 74.2 (2.1) | 76.4 (0.6) | 80.2 (0.7) | 72.80 |
|  | 5 | 70.8 (0.5) | 75.9 (0.5) | 63.1 (0.2) | 60.2 (1.1) | 79.1 (2.8) | 77.7 (0.6) | 76.0 (0.1) | 79.3 (1.5) | 72.76 |
|  | 10 | 70.2 (0.9) | 74.9 (0.4) | 63.4 (0.4) | 60.8 (1.0) | 80.6 (0.4) | 76.4 (2.0) | 77.0 (0.3) | 77.4 (1.3) | 72.59 |
|  | 20 | 69.5 (0.4) | 74.9 (0.4) | 63.3 (0.1) | 60.8 (0.3) | 81.2 (0.5) | 77.3 (2.7) | 76.9 (0.3) | 80.1 (0.5) | 73.00 |
| MVP-C | 1 | 69.7 (0.9) | 74.9 (0.5) | 64.1 (0.5) | 61.0 (1.4) | 78.3 (2.7) | 75.7 (1.5) | 74.7 (0.8) | 81.3 (0.7) | 72.46 |
|  | 5 | 72.4 (1.6) | 74.4 (0.2) | 63.1 (0.4) | 63.9 (1.2) | 77.5 (4.2) | 75.0 (1.0) | 77.0 (1.2) | 81.2 (0.9) | 73.07 |
|  | 10 | 69.5 (1.5) | 74.5 (0.5) | 63.9 (0.9) | 60.9 (0.4) | 81.1 (1.8) | 76.8 (1.5) | 76.0 (0.8) | 82.0 (1.0) | 73.09 |
|  | 20 | 72.1 (0.4) | 73.4 (0.7) | 63.9 (0.3) | 63.0 (0.7) | 78.8 (2.4) | 74.1 (1.0) | 74.8 (0.9) | 84.1 (0.6) | 73.02 |

## G.3 ABLATION STUDY: EFFECT OF EACH LOSS COMPONENT

Table 11: Molecular graph property prediction, we set $C$=5 and $M$=0.15 for GraphMVP methods.

|  | BBBP | Tox21 | ToxCast | Sider | ClinTox | MUV | HIV | Bace | Avg |
|---|---|---|---|---|---|---|---|---|---|
| # Molecules | 2,039 | 7,831 | 8,575 | 1,427 | 1,478 | 93,087 | 41,127 | 1,513 | - |
| # Tasks | 1 | 12 | 617 | 27 | 2 | 17 | 1 | 1 | - |
| - | 65.4(2.4) | 74.9(0.8) | 61.6(1.2) | 58.0(2.4) | 58.8(5.5) | 71.0(2.5) | 75.3(0.5) | 72.6(4.9) | 67.21 |
| InfoNCE only | 68.9(1.2) | 74.2(0.3) | 62.8(0.2) | 59.7(0.7) | 57.8(11.5) | 73.6(1.8) | 76.1(0.6) | 77.6(0.3) | 68.85 |
| EBM-NCE only | 68.0(0.3) | 74.3(0.4) | 62.6(0.3) | 61.3(0.4) | 66.0(6.0) | 73.1(1.6) | 76.4(1.0) | 79.6(1.7) | 70.15 |
| VAE only | 67.6(1.8) | 73.2(0.5) | 61.9(0.4) | 60.5(0.2) | 59.7(1.6) | 78.6(0.7) | 77.4(0.6) | 75.4(2.1) | 69.29 |
| AE only | 70.5(0.4) | 75.0(0.4) | 62.4(0.4) | 61.0(1.4) | 53.8(1.0) | 74.1(2.9) | 76.3(0.5) | 77.9(0.9) | 68.89 |
| InfoNCE + VAE | 69.6(1.1) | 75.4(0.6) | 63.2(0.3) | 59.9(0.4) | 69.3(14.0) | 76.5(1.3) | 76.3(0.2) | 75.2(2.7) | 70.67 |
| EBM-NCE + VAE | 68.5(0.2) | 74.5(0.4) | 62.7(0.1) | 62.3(1.6) | 79.0(2.5) | 75.0(1.4) | 74.8(1.4) | 76.8(1.1) | 71.69 |
| InfoNCE + AE | 65.1(3.1) | 75.4(0.7) | 62.5(0.5) | 59.2(0.6) | 77.2(1.8) | 72.4(1.4) | 75.8(0.6) | 77.1(0.8) | 70.60 |
| EBM-NCE + AE | 69.4(1.0) | 75.2(0.1) | 62.4(0.4) | 61.5(0.9) | 71.1(6.0) | 73.3(0.3) | 75.2(0.6) | 79.3(1.1) | 70.94 |

### G.4 Broader Range of Downstream Tasks: Molecular Property Prediction Prediction

Table 12: Results for four molecular property prediction tasks (regression). For each downstream task, we report the mean (and standard variance) RMSE of 3 seeds with scaffold splitting. For GraphMVP, we set $M = 0.15$ and $C = 5$. The best performance for each task is marked in **bold**.

|  | ESOL | Lipo | Malaria | CEP | Avg |
|---|---|---|---|---|---|
| – | 1.178 (0.044) | 0.744 (0.007) | 1.127 (0.003) | 1.254 (0.030) | 1.07559 |
| AM | 1.112 (0.048) | 0.730 (0.004) | 1.119 (0.014) | 1.256 (0.000) | 1.05419 |
| CP | 1.196 (0.037) | 0.702 (0.020) | 1.101 (0.015) | 1.243 (0.025) | 1.06059 |
| JOAO | 1.120 (0.019) | 0.708 (0.007) | 1.145 (0.010) | 1.293 (0.003) | 1.06631 |
| GraphMVP | 1.091 (0.021) | 0.718 (0.016) | 1.114 (0.013) | 1.236 (0.023) | 1.03968 |
| GraphMVP-G | 1.064 (0.045) | 0.691 (0.013) | 1.106 (0.013) | **1.228 (0.001)** | 1.02214 |
| GraphMVP-C | **1.029 (0.033)** | **0.681** (0.010) | **1.097** (0.017) | 1.244 (0.009) | **1.01283** |

### G.5 Broader Range of Downstream Tasks: Drug-Target Affinity Prediction

Table 13: Results for two drug-target affinity prediction tasks (regression). For each downstream task, we report the mean (and standard variance) MSE of 3 seeds with random splitting. For GraphMVP, we set $M = 0.15$ and $C = 5$. The best performance for each task is marked in **bold**.

|  | Davis | KIBA | Avg |
|---|---|---|---|
|  | 0.286 (0.006) | 0.206 (0.004) | 0.24585 |
| AM | 0.291 (0.007) | 0.203 (0.003) | 0.24730 |
| CP | 0.279 (0.002) | 0.198 (0.004) | 0.23823 |
| JOAO | 0.281 (0.004) | 0.196 (0.005) | 0.23871 |
| GraphMVP | 0.280 (0.005) | 0.178 (0.005) | 0.22860 |
| GraphMVP-G | **0.274 (0.002)** | 0.175 (0.001) | 0.22476 |
| GraphMVP-C | 0.276 (0.004) | **0.168 (0.001)** | **0.22231** |

### G.6 Case Studies

**Shape Analysis (3D Diameter Prediction).** Diameter is an important measure in molecule [60, 62], and genome [22] modelling. Usually, the longer the 2D diameter (longest adjacency path) is, the larger the 3D diameter (largest atomic pairwise l2 distance). However, this is not always true. Therefore, we are particularly interested in using the 2D graph to predict the 3D diameter when the 2D and 3D molecular landscapes are with large differences (as in Figure 2 and Figure 8). We formulate it as a $n$-class recognition problem, where $n$ is the number of class after removing the consecutive intervals. We provide numerical results in Table 14 and more visualisation examples in Figure 9.

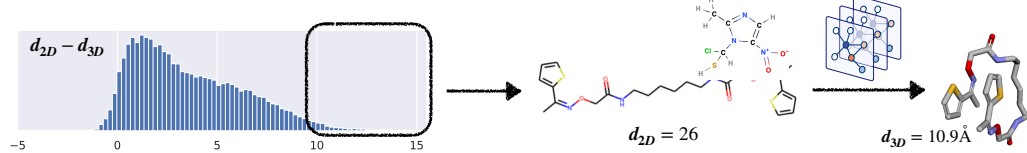

Figure 8: Molecules selection, we select the molecules that lies in the black dash box.

**Long-Range Donor-Acceptor Detection.** Donor-Acceptor structures such as hydrogen bonds have key impacts on the molecular geometrical structures (collinear and coplanarity), and physical properties (melting point, water affinity, viscosity etc.). Usually, atom pairs such as "O...H" that are closed in the Euclidean space are considered as the donor-acceptor structures [46]. On this basis, we are particularly interested in using the 2D graph to recognize (i.e., binary classification) donor-acceptor

Table 14: Accuracy on Recognizing Molecular Spatial Diameters

| Random | AttrMask | ContextPred | GPT-GNN | GraphCL | JOAOv2 | MVP | MVP-G | MVP-C |
|---|---|---|---|---|---|---|---|---|
| 38.9 (0.8) | 37.6 (0.6) | 41.2 (0.7) | 39.2 (1.1) | 38.7 (2.0) | 41.3 (1.2) | 42.3 (1.9) | 41.9 (0.7) | 42.3 (1.3) |

structures which have larger ranges in the 2D adjacency (as shown in Figure 2). Similarly, we select the molecules whose donor-acceptor are close in 3D Euclidean distance but far in the 2D adjacency. We provide numerical results in Table 15. Both tables show that MVP is the MVP :)

Table 15: Accuracy on Recognizing Long-Range Donor-Acceptor Structures

| Random | AttrMask | ContextPred | GPT-GNN | GraphCL | JOAOv2 | MVP | MVP-G | MVP-C |
|---|---|---|---|---|---|---|---|---|
| 77.9 (1.1) | 78.6 (0.3) | 80.0 (0.5) | 77.5 (0.9) | 79.9 (0.7) | 79.2 (1.0) | 80.0 (0.4) | 81.5 (0.4) | 80.7 (0.2) |

**Chirality.** We have also explored other tasks such as predicting the molecular chirality, it is a challenging setting if only 2D molecular graphs are provided [72]. We found that GraphMVP brings negligible improvements due to the model capacity of SchNet. We save this in the ongoing work.

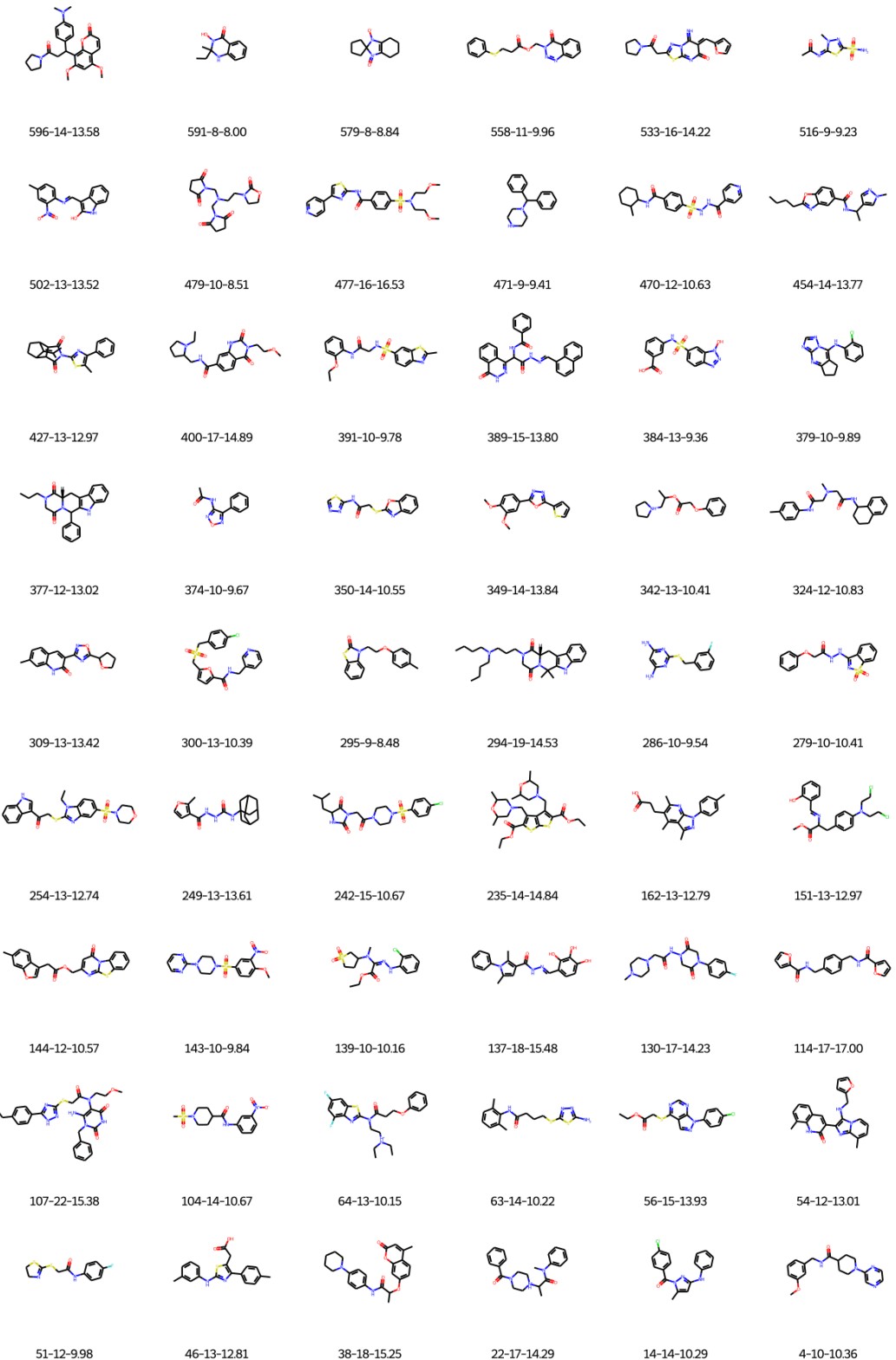

Figure 9: Molecule examples where GraphMVP successfully recognizes the 3D diameters while random initialisation fails, legends are in a format of "molecule id"-"2d diameter"-"3d diameter".

