# OpenReview forum: "Pre-training Molecular Graph Representation with 3D Geometry"
_ICLR.cc/2022/Conference — ICLR 2022 Poster_

### Official Review · Reviewer_K6sw · 2021-10-30

**Correctness:** 4
**Technical Novelty And Significance:** 3
**Empirical Novelty And Significance:** 3
**Recommendation:** 6
**Confidence:** 4

**Main Review:**


This paper proposed GraphMVP to pretrain molecular representations by using 3D information with SSL tasks. Overall, the motivation is strong, and the method is technically sound. My remaining concern is that the experiments could be more convincing by using large datasets and considering tasks that are more dependent on 3D information.

#####Pros#####

1. The intuition of using expensive 3D geometry information to pretrain molecular representation is convincing since (1) 3D geometry is essentially vital for molecular property and (2) it is usually unavailable for downstream tasks. Therefore, the motivation of this work is clear and strong.


2. To achieve the pretraining with considering 3D information, this paper develops one generative and one contrastive SSL task. Although generative and contrastive SSL are widely studied in graph representation learning, extending them into considering both 2D and 3D inputs is non-trivial and well-tackled by this work. One example is that for the generative SSL task this work proposes to do the reconstruction in the latent space instead of data space, where considering reconstruction is much more complicated. This insight is interesting and novel.


3. The ablation studies are extensive and well-presented. The writing and organization of this paper are great.

#####Cons#####

1. The main remaining concern for this paper is that the experiments could be more convincing. The current results are mostly on small datasets (See Table 8 in supplementary) and it is not clear that if the included tasks are highly related to 3D information. I suggest the authors to consider larger datasets and more convincing tasks. The recent OGB KDD Cup task (https://ogb.stanford.edu/kddcup2021/) could be a good candidate. The task is to predict the quantum chemical property, which is highly dependent on the 3D geometry.


#####Questions#####

1. In Section 3.1, it is mentioned that the contrastive SSL learns the distribution locally while the generative SSL learns the global distribution. This description is confusing to me. More explanations of this point should be considered.


**Summary Of The Paper:**

This paper proposes to leverage 3D information to pretrain graph neural networks for learning molecular representations. To achieve this, contrastive and generative SSL strategies are developed accordingly. The empirical results on several datasets show that the improvements of this method over previous 2D graph pretraining methods are consistent and obvious.

**Summary Of The Review:**

The current version of this paper is good and above the acceptance threshold. More convincing experiments could improve the quality a lot. Hence, I recommend a weak accept to the current version and will consider improving the score if the experiments could be improved significantly.

---

> ### Author Response · Authors · 2021-11-12
> **Larger downstream dataset requires larger pre-training dataset**
>
> First we appreciate the reviewer for the insightful comments. Below are our answers.
>
> 1. About dataset. We actually considered applying GraphMVP on PCQM4M for downstream, which is more related to the 3D information. We finally abandoned it for mainly two reasons.
>     - For SSL pre-training, or more general pre-training setting, we are always expecting the pre-training dataset is large, at least orders of magnitude larger than the downstream tasks. Now GEOM has only 400k molecules, but PCQM4M has ~3.7M molecules, and we can’t find any other clean molecule datasets with both over 10M molecules and 3D information.
>     - Some top-ranked solutions for PCQM4M are actually generating 3D conformers in a heuristic way, where it would be unfair to compare with GraphMVP, since we assume only 2D is available for downstream tasks.
>
>     - To make our results more robust, in addition to the 8 widely-used tasks in Table 1, we added 6 extra tasks in Table 5. We didn’t do too much tuning, yet, the performance improvement is **consistent**. Such consistency helps support the effectiveness of GraphMVP. In addition, we do agree with the reviewer that this is an interesting direction, just that we may need to get a more comprehensive pre-training dataset. This may need efforts from the whole community.
>
> 2. About the question. This is actually a very insightful point, and we appreciate the reviewer for carefully noticing this. Below are more explanations.
>     - First, in our work, we formulate both the contrastive and generative SSL as optimizing Eq 9, but with different methods.
>     - Contrastive SSL methods optimize Eq 9  by doing alignment and contrasting simultaneously. Some recent work has explored how pairwise contrasting can lead to a well-separated representation space, like Figure 3 in [a]. We can think of this as contrasting each pairwise data point, and as we keep training, such local pairwise contrasting can gradually lead to the widely spread representation space according to certain hidden structure/label (again, like the Figure 3 in [a]).
>     - Generative SSL methods, on the other hand, do not consider local contrasting, and each data point is used to model the conditional likelihood directly. To better understand this, we can imagine that in VAE, each individual data point is used to shape the mode of the whole distribution directly. Thus, we term such direct approximation as learning the global distribution.
>     - We find this point quite insightful, but actually hasn’t attracted much attention in the literature. The terms `local` and `global` might not be exact, and we are working on a more rigorous definition. We appreciate the reviewer for asking this, and hope this helps answer your question.
>
> [a] Wang, Tongzhou, and Phillip Isola. "Understanding contrastive representation learning through alignment and uniformity on the hypersphere." International Conference on Machine Learning. PMLR, 2020.

---

> > ### Comment · Reviewer_K6sw · 2021-11-18
> > **Thanks for the clarification**
> >
> > Thank you for answering the raised questions. I am currently having no other questions and will discuss these with other reviewers and ACs.

---

### Official Review · Reviewer_A6B1 · 2021-11-02

**Correctness:** 4
**Technical Novelty And Significance:** 3
**Empirical Novelty And Significance:** 2
**Recommendation:** 6
**Confidence:** 3

**Main Review:**

The strengths are :
1. The idea of using 3D information to augment the 2D graph representation learning is interesting and novel. Indeed, 3D geometric information is useful and provides a complementary view of the data, as evidenced in existing works such as protein classification.
2. Fusing contrastive and generative losses can empirically further improve the performances by providing both inter-data (local) and intra-data (global) viewpoints.
3. Extensive experiments were conducted (e.g., two tasks, a number of recent baselines, and 14 datasets are involved). Via the experiments, we can see that the proposed method shows promising performances across various datasets. Some useful insights are given in the ablation study.
4. The paper is well-written and easy to follow.

Yet, there are also some weaknesses:
1. My major concern is the fairness of the comparison. I am afraid that the major improvement is brought by simply incorporating 3D information as additional features, which raises a question:  If other baseline methods have 3D information as additional features, will they perform better (or even better than GraphMVP)? If so, the contribution might be weakened.
2. It seems that x and y notations are flipped. In preliminary sections, x denotes 2D and y denotes 3D; In the method section, x denotes 3D and y denotes 2D. This is a little bit confusing to readers.
3. The improvements on some datasets seem to be marginal. More results analyses are required.

Minor comments:
1. Typo: and are different (page 4).
2. The appendix can be condensed.



**Summary Of The Paper:**

This work aims to leverage additional 3D geometric information for molecular graph representation learning and proposes a multi-view pre-training framework, GraphMVP. Specifically, in GraphMVP, both 2D and 3D information are used, and a combined loss function (i.e., the combination of contrastive self-supervised learning loss and generative self-supervised learning loss ) is adopted to enhance the quality of representations. To verify the model's effectiveness, they conducted experiments and comparisons on a number of datasets. Ablation studies are also carried out to inspect the inner workings of GraphMVP.

**Summary Of The Review:**

This paper proposes integrating 3D geometric information to improve molecule graph representation learning. The idea is interesting and the experiments are comprehensive.

---

> ### Author Response · Authors · 2021-11-12
> **Fair comparison and consistent improvements**
>
> First we appreciate the reviewer for the insightful comments. Below are our answers.
>
> 1. `I am afraid that the major improvement is brought by simply incorporating 3D information as additional features…`.
>     - We want to clarify that, for both the pre-training and fine-tuning, we did not change the input feature to the 2D GNN (in Eq 1), and it is exactly the same as other SSL baselines.
>     - We are not adding any extra 3D information into the 2D GNN, otherwise it would be a biased comparison. To be more specific, the only thing that has been changed in GraphMVP with other SSL is how to get the 2D representation from the pre-training stage. All the fine-tunings, 2D features, and 2D GNN models stay the same.
>     - We also want to highlight that our setting is: the downstream tasks do not have 3D information, thus these extra features are not available either.
>     - Thus, this is a fair comparison. The consistent performance gain comes from a better pre-trained 2D representation, i.e., it reveals the success of our proposed GraphMVP algorithm and can support the effectiveness of GraphMVP.
>
>
> 2. Thanks for pointing this out. x should be 2D, and y should be 3D. ~~We found one mismatch after Eq 3. We will do more proof-reads and fix them in the final version.~~ We have fixed three mismatched places in Sec 3.2 in the latest revision.
>
> 3. We want to highlight that in this research literature, these tasks are well-studied and well-tuned, thus they are actually very hard to improve, especially when comparing with other SSL methods. The latest result before our work is JOAO [a], and the improvements are quite small. The reviewer can check Table 6 in [a]. While in our work, we didn’t do too much hyper-parameter tuning, yet the performance is **consistently** better (see Table 1, Table 5 in our submission), plus the better performance gain. This consistency, especially when comparing with other SSL methods, verifies the effectiveness of GraphMVP.
>
> 4. For minor comments. Thanks for pointing them out, and we will fix them in the final version.
>
> [a] You, Yuning, et al. "Graph Contrastive Learning Automated." arXiv preprint arXiv:2106.07594 (2021).

---

### Official Review · Reviewer_1JZt · 2021-11-02

**Correctness:** 3
**Technical Novelty And Significance:** 2
**Empirical Novelty And Significance:** 2
**Recommendation:** 5
**Confidence:** 4

**Main Review:**

The paper is well written and easy to follow. The focused problem of taking into account 3D information in GNN pretraining is also very interesting and important. The idea sounds interesting, but at the same time, a big question on how to handle "conformer ensemble" seems not well addressed. 3D information is not unique for a single molecule, and this would be the main difficulty why past work didn't take into account it well. For now, it would be a bit unconvincing that this method worked as intended.

So a big natural question is how this method could handle multiple 3D poses of a molecule. Just all of the possible 3D conformers fed into learning by data augmentation? As the paper also mentioned, "the molecular properties are a function of the conformer ensembles", and 3D information is not unique for a given molecule. But it is unclear how to address this ensemble problem. In theory, a molecule has infinitely many 3D shapes in most cases, because any single carbon-carbon bond can be rotated to any degree. We can continuously rotate every C-C bond to produce massive sets of 3D geometries? How to handle them? Or pick up locally energy-stable ones in some ways...? This point seems to remain unclear. Also, the datasets in experiments mainly come from tasks predicting the bioactivities of molecules, and molecules would have a huge degree of such degrees, as well as actual 3D shape in action, might be determined by other interacting molecules (such as target proteins) and also statistical-mechanically fluctuated continuously, and so it would be unconvincing that learning only a few of stable 3D conformations is sufficient.

For your interest: Bio QSAR tasks would require some handling of the combinatorial multiplicity of 3D conformations of a molecule when we try to include 3D information (typically by docking simulations or something?). I guess that's why (2D) GNNs are sufficient usually. We can ignore complicated problems by continuous/statistical-mechannical 3D confomations and their dynamics and abstract a molecule as a graph (dots and lines) to see statistical trends. This method might be good for a quantum chemical task like QM9 where we are trying to predict the energy (or properties) of a molecule with a specified 3D conformation (an energetically stable structure).



**Summary Of The Paper:**

This paper presents a pre-training method of GNNs for molecular graphs by self-supervised learning (SSL) on not only 2D topologies of graphs but also 3D geometries of molecules.  In particular, the paper develops two SSL pretext tasks learning inter-molecule and intra-molecule associations in 2D and 3D. The "inter-molecule" task is a graph classification on whether a 2D, 3D graph pairs come from the same molecule or not, while the "intra-molecule" task is a generative task to generate 2D from 3D as well as 3D from 2D information. Empirical evaluations over several datasets demonstrate that jointly learning these two SSL tasks brings nice improvement in prediction accuracies.

**Summary Of The Review:**

The paper is well written and easy to follow. It provides interesting and effective pretraining methods for GNNs. However, it is still not clear how to handle the combinatorial multiplicity of 3D molecular information in a principled way even though we can observe empirical improvement in prediction accuracies by this pretraining in a limited set of datasets.

---

> ### Author Response · Authors · 2021-11-12
> **Clarification on how GraphMVP handles multiple conformers and clarification on downstream datasets (3/3)**
>
> ### 3 Responses on the last paragraph
>
> The reviewer mentions certain points in the last paragraph, but some of them reveal that this reviewer misunderstood our work, and we may as well list them below.
>
> 1. `I guess that’s why 2D GNNs are sufficient usually.`
>     - We are wondering if the reviewer has any reference to support this claim? Several works [c] have proven this is not the case for quantum mechanics tasks; but we haven't’ seen any direct proof or falsification about `2D info is sufficient` from other tasks yet (physiology, physical chemistry, biophysics; or what the reviewer called Bio QSAR).
>     - Our work is motivated by the conjecture that, `3D conformer can be helpful`. This point has already been well acknowledged by reviewer zY9H, A6B1, K6sw as convincing and promising. Besides, our **empirical performance can strongly support this conjecture**. The performance gain from our work (average ROC from 67.21 to 73.07 in Table 1, or the RMSE/MSE reduction in Table 5) can again prove our conjecture: 3D information can indeed help these downstream tasks. This falsifies the reviewer's guess.
>
> 2. `This method might be good for quantum chemical tasks like qm9 … of a molecule with a specified 3D conformation.` We believe that the reviewer may misunderstand the fundamental setting of our work.
>     - As mentioned above, GraphMVP can handle multiple conformers.
>     - In our setting, what we focus on is how to **pre-train with 3D info to help the downstream tasks without 3D information**. We highlight this in Abstract, Sec 1, Sec 3, and Sec 5. Thus, QM9 is definitely not suitable since it has 3D info; and apparently, the reviewer misses this basic setting.
>
> ### 4 Summary
>
> To sum up, this reviewer has 2 main concerns which are already solved in the original submission, and our rebuttal clarifies the main points again. We are happy to answer any additional questions.
>
> ----
>
> [a] Wu, Zhenqin, et al. "MoleculeNet: a benchmark for molecular machine learning." Chemical science 9.2 (2018): 513-530.
>
> [b] Axelrod, Simon, and Rafael Gomez-Bombarelli. "GEOM: Energy-annotated molecular conformations for property prediction and molecular generation." arXiv preprint arXiv:2006.05531 (2020).
>
> [c] Gilmer, Justin, et al. "Neural message passing for quantum chemistry." International conference on machine learning. PMLR, 2017.

---

> > ### Comment · Reviewer_1JZt · 2021-11-19
> > **thanks**
> >
> > > 1
> >
> > This doesn't mean to deny the claim of this work. I agree that adding 3D information in SSL pretext tasks empirically improved the performance at 2D downstream tasks. And that's good because 2D info is often available in downstream tasks. Here I just tried to make sure why only 2D info is usually used for MoleculeNet-like tasks because 2D info is readily available whereas 3D info has representation difficulties due to the conformations. A famous example would be the ChemProp's success in actually discovering real antibiotics using exploration assisted by a 2D GNN. (https://doi.org/10.1016/j.cell.2020.01.021)
> >
> > > 2
> >
> > Yes, this is my misunderstanding. Quantum chemical datasets have one-to-one correspondences between geometries and targets, and so I just thought that it can avoid this difficult 3D conformational problem.

---

> > > ### Author Response · Authors · 2021-11-20
> > > **Follow-up replies**
> > >
> > > First we appreciate the reviewer for carefully reading our paper and replies, and we are very glad that part of the concerns have been addressed.
> > >
> > > We realize that our paper may contain too much content, and some sentences can be easily missed by the readers. We appreciate the reviewer for raising questions, as we will further clarify below.
> > >
> > > > But my original question still partly remains …
> > > - Our apologies for not making that clear in the original text.  Sec F is showing the statistics of the GEOM dataset. Then we find that in GEOM, over 86% of molecules can be well represented (the sum of Boltzmann weights are larger than 90%) with less than 20 conformers (Figure 7), thus, we only take top-C (c=5) conformers for each molecule. We implicitly mentioned this in Sec 4.1, and explicitly in Sec F, and further supported with an ablation study on the effect of C (c=1,5,10,20) in Sec 4.3. We also briefly mentioned this in the first round rebuttal (see point 4 in (1/3) and point 2 in (2/3)). Please feel free to check more details there and hopefully the texts now are much clearer to the reviewer.
> > >
> > > > If you clearly distinguished "conformers" and "conformations"...
> > > - First we want to clarify that GraphMVP does two directions, both from 2D to 3D and 3D to 2D generation.
> > > - Second, we want to point out that we realize the generation issue, including conformer generation (in the second paragraph in Sec 3.3), and that’s why our final solution is to move the reconstruction from data space to representation space, i.e., variational representation reconstruction (VRR). Our VRR in fact aims to avoid the 2D/3D reconstruction issue as asked by the reviewer. We appreciate the reviewer for raising this question, which indeed highlights one of our paper’s main contributions.
> > >
> > > > Could you somehow …
> > > - Thank you for the insightful question. We do not use or generate any type of 3D information for downstream tasks, which is consistent with our setting: no 3D information is available for fine-tuning. The only thing that has been transferred from pre-training to fine-tuning, is the pre-trained 2D GNN.
> > > - This is actually the same concern with reviewer A6B1, and we provide more detailed discussions there.
> > >
> > > ---
> > > For the other questions, we are glad that we made the points clear to the reviewer and we appreciate the reviewer for sharing the relevant paper. We will add them in the final version for more comprehensive discussion. We are happy to answer any additional questions.
> > > We would also like to reiterate how grateful we are to reviewer 1JZt for promptly replying to our rebuttal.

---

> > > > ### Comment · Reviewer_1JZt · 2021-11-25
> > > > **Thanks for the response!**
> > > >
> > > > I acknowledge that I have read and understood the points. In particular, the VRR loss is defined over the (latent) representation space, and minimizing it brings us a good latent representation through both pretext tasks of 2D to 3D and 3D to 2D, and molecular reconstruction itself is not a direct purpose here. And the paper empirically demonstrated that these learned latent features were good in the sense that they improved performance in multiple downstream tasks.
> > > >
> > > > I have no further questions, and would like to discuss the given contributions with other reviewers and ACs.

---

> > > > > ### Author Response · Authors · 2021-11-25
> > > > > **Appreciate your time for the helpful discussion!**
> > > > >
> > > > > Thank you again for taking the time to discuss the paper with us. We sincerely appreciate the very helpful discussion,
> > > > > and would welcome any ideas now, or after the review period! At the same time, if we have addressed your concerns,
> > > > > we humbly ask if you would kindly consider supporting accepting the paper during the discussion
> > > > > with other reviewers/ACs or by increasing the review score accordingly.

---

> ### Author Response · Authors · 2021-11-12
> **Clarification on how GraphMVP handles multiple conformers and clarification on downstream datasets (2/3)**
>
> ### 2 Responses on the dataset
> 1. ` … the datasets in experiments mainly come from tasks predicting the bioactivities of molecules ...`
>
>     - With all due respect, we disagree on this claim, and the datasets used in our study are not only about bioactivities (explained below), we would like to see and response if there is specific point/reference provided. Following the categories of [a], tasks in our study can be mainly classified as four types (only first three are used):
>       - physiology: Tox21, ToxCast, ClinTox, BBBP, Sider
>       - physical chemistry: ESOL, Lipophilicity, CEP (newly added)
>       - biophysics: MUV, BACE, Hiv, Malaria (newly added)
>        - quantum mechanics: QM7, QM8, QM9 (not fit in the GraphMVP setting, since we have no 3D information in downstream tasks)
>     - Thus, we have shown that our downstream tasks actually cover a broad range of tasks. Besides, all these tasks have been widely used in the **SSL literatures** [31, 71, 38, 87, 66, 90, 89].
>
> 2. `so it would be unconvincing that learning only a few of stable 3D conformations is sufficient`.
>     - First, we want to highlight that we haven’t drawn this conclusion in our submission.
>     - Second, we only have conformers in the pre-training step. At pre-training, we use GEOM [b], where over 86% of molecules can be well represented (the sum of Boltzmann weights are larger than 90%) with less than 20 conformers. For more details, please check [b] or Sec F.1.
>     - Third, we **explicitly** take an ablation study to check the effect of the number of conformers in Sec 4.3 and Table 3. Our conclusion is that, from the algorithm design point of view, `we would encourage tuning masking ratios prior to trying a larger number of conformers.`
>
> 3.  `... as well as actual 3D shape in action, might be determined by other interacting molecules and also statistical-mechanically fluctuated continuously ...`
>     - This is actually constrained by the pre-training dataset. If there exists a large-scale dataset that also provides how each molecule conformation changes with the interaction to the target proteins, then we would definitely use it.
>     - But **at the current stage**, we haven’t seen any work on this, and GraphMVP is the **first** to incorporate 3D info for 2D pre-training.
>
> 4. We don’t quite get what the reviewer meant by `... molecules would have a huge degree of such degrees`. Can the reviewer provide any further explanation?

---

> > ### Comment · Reviewer_1JZt · 2021-11-19
> > **thanks**
> >
> > > 1
> >
> > Sorry, my original comment was not clear enough, I meant that molecules are of the size in biological research. So this point is well taken. Molecules of these sizes can have a large number of rotatable bonds, and I tried to make sure that the number of conformers/conformations is large too. But as the author pointed out, Sec F.1 also made this clear. So my question was cleared.
> >
> > > 2, 3
> >
> > This point is well taken. Thanks for the clarification.
> >
> > > 4
> >
> > As mentioned before, my primary questions of how good one-to-many correspondence between x and y can be handled in the VAE-like treatment. So this statement just means "molecules in GEOM/MoleculeNet have a large number of rotatable bonds".

---

> ### Author Response · Authors · 2021-11-12
> **Clarification on how GraphMVP handles multiple conformers and clarification on downstream datasets (1/3)**
>
> First, we appreciate the reviewer for carefully reading our paper.
>
> Briefly, the reviewer has two concerns: (1) GraphMVP doesn’t capture the “conformer ensemble”, and (2) the datasets are all about bioactivities. With all due respect, we disagree on both points, since **both ‘concerns’ are already explicitly addressed in our submission** yet unfortunately missed by the reviewer. We again emphasize the main points as follows, where each point is supported with numbers/references/experiments.
>
> ### 1 Responses on the conformer ensemble
>
> First, it is worth mentioning that the concepts of conformer and conformation are **fundamentally different** in stereochemistry. The conformer refers to the stable 3D structure that exists in nature, while the conformation corresponds to any possible 3D structures of a molecule. Take butane as an example, its Relative conformation energy as a function of dihedral angle is plotted in this diagram. While there are an infinite number of conformations yet only three conformers. Since we notice the reviewer might mix these two concepts, we make these clarifications to mitigate the understanding gaps.
>
> In other words, we are aware that each molecule corresponds to several (sometimes thousands of) 3D structures (ensemble of conformers). Therefore, **we have carefully handled this issue in pre-training and pre-processing (Sec 3) and ablation studies (Sec 4).** We elaborate more details as follows.
>
> 1. **The contrastive SSL has avoided this issue implicitly**, as long as we are creating positive and negative pairs following Eq 3 and 4. Namely, for each 2D molecular graph x, it has multiple 3D counterparts, and each of them can compose a positive pair with x. BTW, this also explains why EBM-NCE is better than InfoNCE, because EBM-NCE only considers positive and negative pairs directly, while InfoNCE still has a normalization of negative pairs in the current batch.
>
> 2. **For the generative SSL, we explicitly addressed this**. Since the mapping between 2D and 3D views is not deterministic, we would like to consider the VAE-like framework, which leads to our novel structure-oriented generative SSL, i.e., the Variational Representation Reconstruction (VRR) in Eq 6. Please carefully read the first two paragraphs in Sec 3.3.
>
> 3. **Further in Sec 4.4, we also carry out an ablation study to explicitly verify that randomness in VRR can better capture the conformer information**. We also explicitly draw this conclusion in the last sentence in Sec 4.4.
>
> 4. **The preprocessing of the pre-trained dataset, GEOM [b], also handles this conformer ensemble issue**, and we provide some statistics to justify it in Sec F.1. In GEOM, there is an additional attribute of each conformer called Boltzmann Weight, which essentially corresponds to their expected existence probabilities at the natural in-situ environment. We notice that over 86% of molecules can be well represented (the sum of Boltzmann weights are larger than 90%) with less than 20 conformers. So in experiments, we start by selecting the more representative (i.e., with largest weights) conformers for the pre-training.
> On this basis, we further explore how adding more conformers affects the improvements/effects in the ablation studies. For the downstream tasks, we do not have this issue since only 2D graphs are available.

---

> > ### Comment · Reviewer_1JZt · 2021-11-19
> > **Thanks for the responses!**
> >
> > Thank you for the detailed response. As the authors pointed out, I didn't distinguish conformers from conformations since the cited GEOM paper used the word conformations. Now I understand the difference and am convinced why the VAE-like approach was taken, and so I'm going to raise my rating score from 3 to 5.
> >
> > But my original question still partly remains because I assumed the point 1 "for each 2D molecular graph x, it has multiple 3D counterparts, and each of them can compose a positive pair with x." As Sec F.1 showed, this can be a large number, isn't it..?
> >
> > If you clearly distinguished "conformers" and "conformations", then what was actually learned through this ill-posed inverse setting of the SSL task from 2D to 3D..? VAE is a probabilistic treatment so this is still unclear. Did you check what y are typically output from a specific x?? Is it similar to one of the conformers or something like averaged conformations...?? Would this be more like vaguely averaged "conformations" than individual "conformers"..?
> >
> > Any learned NN cannot generate a correct 3D info (y) from 2D info (x) because we have multiple correct answers as y. If it can return one of the correct answers, then this means it fails to predict all other correct answers.
> >
> > Could you somehow make clear that the observed improvement is not just simply from adding more information (3D info originally not available in downstream tasks) to solve downstream 2D tasks..?

---

### Official Review · Reviewer_zY9H · 2021-11-04

**Correctness:** 3
**Technical Novelty And Significance:** 3
**Empirical Novelty And Significance:** 3
**Recommendation:** 5
**Confidence:** 4

**Main Review:**

For molecular structure, it is natural and necessary to consider 3D geometry information together with graph structure. This would be a great direction to explore. The proposed method makes sense.

 My concern is the experimental part where only molecular property prediction has been discussed. It would be necessary to illustrate the proposed graph and 3D geometry method can indeed improve reconstruction and generation for molecular data by comparing with other state-of-the-art graph-based methods.

**Summary Of The Paper:**

The manuscript proposes a method for molecular graph representation learning by combing graph topological structure with 3D geometric information.

**Summary Of The Review:**

I like the proposed idea. However, some experiments on reconstitution and generation would better illustrative the improvements using the proposed method.

---

> ### Author Response · Authors · 2021-11-11
> **Generation downstream tasks are out of the scope of SSL pre-training**
>
> Thanks for spending the time on our submission. We deeply appreciate the feedback. We hope our rebuttal can clarify your concerns
>
> We respectfully disagree with the suggestion that using generation for downstream tasks. This involves some basic understandings/clarifications of self-supervised (unsupervised) pre-training, and we listed 2 main points below.
>
> (1) **The motivation of SSL or unsupervised pre-training does not match that of generation downstream tasks.** Recall that the core motivation of SSL pre-training is to learn a good representation with a large-scale dataset, and then fine-tune it on downstream prediction tasks, whose **ground-truth labels are usually insufficient**.
> 1. This is appealing because downstream prediction tasks often do not have enough ground-truth labels, and pre-training with different inductive biases can lead to better initialized representation.
> 2. As asked, what if the downstream is generation tasks? For generation or density estimation tasks, we do not require the ground-truth labels. In other words, the label insufficiency is a bottleneck for prediction rather than generation. Thus, the primary motivation for SSL **does not relate to** generation.
> 3. If we have a large unlabeled dataset, we can learn the data distribution on it directly, and there’s no requirement for the two stages (pre-training + fine-tuning). At least in the literature of SSL pre-training, we do not manage to find any motivation for that. (Also none of the related work supports this, as will be explained below.) We sincerely appreciate it if the reviewer could provide a few relevant literatures.
>
> (2) **None** of the previous/present SSL pre-training work considers the generation for downstream tasks.
> 1. For general setting, past and present SSL methods like Deep InfoMax, SimCLR, SimCLRv2, BYOL, SimSiam, VICReg, none of them is using generation for downstream tasks.
> 2. For molecule and graph SSL methods, we discuss 10 mainstream methods [31, 71, 39, 38, 87, 66, 90, 89] (citations from the submission paper) in Sec 1 and Sec A, and none of them considers generation for downstream tasks.
> 3. The survey papers [48, 50, 83, 85] (citations from the submission paper) show over 100 papers on SSL, and none of them is doing generation as downstream tasks.
>
> As our work is to show that using 3D information helps 2D representation learning through Graph SSL pre-training (GraphMVP), downstream tasks such as generation are beyond the scope of this paper.

---

### Comment · Area_Chair_8RQm · 2021-11-20
**AC Discussion**

Dear Reviewers,

Thank you for responding to author feedback. We need more information to move forward. Could all of you read author rebuttals and other review comments to see if anything new to you? Please also raised further questions if you need more information to make a recommendation.

Thank you!

ICLR AC

---

### Comment · Area_Chair_8RQm · 2021-11-24
**AC Discussion**

Reviewer zY9H and Reviewer 1JZt: It seems you two are concerned with this work. I see that authors have provided responses to your concerns. Could you please check and respond?

Reviewer A6B1: There are outstanding responses from authors to you that are not reacted. Could you please do?


Reviewer K6sw: Thank you!

---

### Author Response · Authors · 2021-11-29
**Any questions before the end of the discussion period?**

We would like to thank again all reviewers.

Please let us know if there are additional questions or concerns before the end of the discussion period. We would be happy to discuss or address any additional comments.

---

### Decision · Program_Chairs · 2022-01-20

**Decision:**

Accept (Poster)

**Comment:**

This paper studies the problem of how to use 3D molecular geometry information during training to improve performance during prediction time when 3D information is not available. This is a highly interesting problem as obtaining 3D molecular geometry information requires expensive calculations and such information is usually not available in practice during prediction, while there are some training data with both 2D and 3D information. The work proposes to use self-supervised, predictive and generative approaches to make use of such information. The reviewers overall expressed mixed recommendations. One of the reviewers who scored 5 did not provide further feedback after author response even being prompted multiple times. The other reviewer who scored 5 actively participated in discussion and it seems most of the concerns have been addressed. Given the importance of this problem, and this work seems to be among the first to address this problem, I lean toward accept.